# SIRT2 deacetylase regulates the activity of GSK3 isoforms independent of inhibitory phosphorylation

Mohsen Sarikhani[1], Sneha Mishra[1], Sangeeta Maity[1], Chaithanya Kotyada[2], Donald Wolfgeher[3], Mahesh P Gupta[4], Mahavir Singh[2], Nagalingam R Sundaresan[1]*

[1]Department of Microbiology and Cell Biology, Indian Institute of Science, Bengaluru, India; [2]Molecular Biophysics Unit, Indian Institute of Science, Bengaluru, India; [3]Department of Molecular Genetics and Cell biology, University of Chicago, Chicago, United States; [4]Department of Surgery, University of Chicago, Chicago, United States

**Abstract** Glycogen synthase kinase 3 (GSK3) is a critical regulator of diverse cellular functions involved in the maintenance of structure and function. Enzymatic activity of GSK3 is inhibited by N-terminal serine phosphorylation. However, alternate post-translational mechanism(s) responsible for GSK3 inactivation are not characterized. Here, we report that GSK3$\alpha$ and GSK3$\beta$ are acetylated at Lys246 and Lys183, respectively. Molecular modeling and/or molecular dynamics simulations indicate that acetylation of GSK3 isoforms would hinder both the adenosine binding and prevent stable interactions of the negatively charged phosphates. We found that SIRT2 deacetylates GSK3$\beta$, and thus enhances its binding to ATP. Interestingly, the reduced activity of GSK3$\beta$ is associated with lysine acetylation, but not with phosphorylation at Ser9 in hearts of SIRT2-deficient mice. Moreover, GSK3 is required for the anti-hypertrophic function of SIRT2 in cardiomyocytes. Overall, our study identified lysine acetylation as a novel post-translational modification regulating GSK3 activity.
DOI: https://doi.org/10.7554/eLife.32952.001

*For correspondence:
rsundaresan@iisc.ac.in

**Competing interests:** The authors declare that no competing interests exist.

## Introduction

Glycogen synthase kinase 3 (GSK3) is a highly conserved, ubiquitously expressing kinase having two isoforms, $\alpha$ and $\beta$ (*Woodgett, 1990*). Both these GSK3 isoforms are encoded by distinct genes, which share 97% amino acid sequence identity within their catalytic domains and differ significantly outside the catalytic domain (*Kaidanovich-Beilin and Woodgett, 2011*). In cardiomyocytes, GSK3$\alpha$ preferentially localizes inside the nucleus, while GSK3$\beta$ is present mostly in the cytoplasm (*Matsuda et al., 2008*). Increasing lines of evidence suggest that GSK3$\alpha$ and GSK3$\beta$ have both common and non-overlapping functions (*Rayasam et al., 2009*). Unlike other kinases, GSK3 requires a primed substrate for its function (*Cohen and Frame, 2001*; *Doble and Woodgett, 2003*). Activity of GSK3 is carefully regulated by the combination of (a) Tyrosine phosphorylation at catalytic domain (*Dajani et al., 2001*; *Hughes et al., 1993*), (b) N-terminal serine phosphorylation (*Cross et al., 1995*) and (c) sub-cellular localization (*Beurel et al., 2015*). Studies indicate that GSK3 is constitutively phosphorylated (Tyr279-GSK3$\alpha$ and Tyr216-GSK3$\beta$) and that tyrosine phosphorylation is necessary for its catalytic activity. Phosphorylation of Tyr279 in GSK3$\alpha$ and Tyr216 in GSK3$\beta$ has been reported to facilitate GSK3 activity by promoting substrate accessibility (*Dajani et al., 2001*; *Hughes et al., 1993*). Interestingly, phosphorylation of GSK3$\beta$ at Tyr216 occurs through intramolecular autophosphorylation during protein folding (*Cole et al., 2004*). In contrast, N-terminal serine

phosphorylation of GSK3 that is Ser21- GSK3α and Ser9-GSK3β by several kinases including Akt, protein kinase A, protein kinase C and p90Rsk has been shown to decrease its catalytic activity (*Ali et al., 2001*; *Beurel et al., 2015*; *Cross et al., 1995*; *Song et al., 2015*). GSK3β is inhibited by N-terminal Ser9 phosphorylation, in which the phosphorylated N terminus binds as a competitive pseudo-substrate with p-Ser9 occupying the p+4 site (*Dajani et al., 2001*; *Stamos et al., 2014*). Previous studies have shown that GSK3β-S9A mutation, which prevents the N-terminal Ser9 phosphorylation, causes GSK3β to be constitutively active (*MacAulay et al., 2005*; *Stambolic and Woodgett, 1994*). GSK3β was also observed to be inhibited by phosphorylation at Ser389 by p38 MAPK (*Thornton et al., 2008*). However, alternate post-translational mechanism(s) responsible for the GSK3 inactivation are not yet characterized.

Reversible lysine acetylation is known to regulate nuclear and non-nuclear proteins in the cell (*Verdin and Ott, 2015*). The acetylation reaction is catalyzed by a group of enzymes called histone acetyltransferases and is reversible by histone deacetylases (HDACs) (*Haberland et al., 2009*). Reversible acetylation has been shown to be responsible for activation/inactivation, sub-cellular localization, DNA binding, protein-protein interaction, membrane trafficking, stability and degradation of proteins (*Choudhary et al., 2009*; *Jeffers and Sullivan, 2012*; *Yang and Seto, 2008*). Acetylation of a protein may enhance or repress its activity, and it has been shown that acetylation competes with phosphorylation (*Kouzarides, 2000*), and interferes with other lysine modifications, ubiquitination or sumoylation (*Yang and Seto, 2008*). HDACs are classified into three distinct groups based on their structure and function (Class I to III) (*Haigis and Guarente, 2006*). The sirtuins, which are class III HDACs, are catalytically distinct, as they require $NAD^+$ for their deacetylase activity. In mammals, seven different sirtuin isoforms (SIRT1 to 7), have been characterized (*Guarente, 2007*; *Longo and Kennedy, 2006*). SIRT2 is the cytoplasmic deacetylase that affects the microtubule network by deacetylating α-tubulin (*North et al., 2003*). In addition, SIRT2 regulates nuclear envelope dynamics, cardiac hypertrophy, metabolism and stress-induced cell death (*de Oliveira et al., 2012*; *Kaufmann et al., 2016*; *Sarikhani et al., 2018a*; *Sarikhani et al., 2018b*). In neurons, inhibition of SIRT2 salvage α-synuclein toxicity in Parkinson's disease and reduce sterol-mediated toxicity in Huntington's disease (*Donmez and Outeiro, 2013*). In contrast, SIRT2-deficient mice show increased mammary tumors, hepatocellular carcinoma and heart failure (*Kim et al., 2011*; *Tang et al., 2017*). Similarly, SIRT2 deficiency increases susceptibility to colitis and iron-deficiency-induced hepatocyte cell death (*Lo Sasso et al., 2014*; *Yang et al., 2017*).

In the present work, we demonstrate that SIRT2 binds to and deacetylates GSK3 isoforms to promote ATP binding. Interestingly, reversible acetylation regulates the activity of GSK3 isoforms independent of its inhibitory phosphorylation. We also demonstrate that activity of GSK3 isoforms are essential for inhibition of cardiac hypertrophy by SIRT2 deacetylase.

## Results

### Acetyltransferase p300 regulates the GSK3β acetylation

To test whether GSK3β acetylation plays a role in the development of heart failure, we first created a mice model of cardiac hypertrophy. The cardiac hypertrophic agonist, isoproterenol (ISO) was infused chronically into mice by implanting osmotic mini-pumps in the peritoneal cavity (*Sundaresan et al., 2009*; *Sundaresan et al., 2012*). Chronic ISO infusion significantly increased heart weight/tibia length (HW/TL) ratio as well as left ventricular wall thickness, which is an indicator of cardiac hypertrophy, while reducing the ejection fraction of heart (*Figure 1A–C*), suggesting that ISO-infusion causes adverse cardiac remodeling and contractile dysfunction in mice. Previous studies indicate that GSK3β activity is reduced in hypertrophic hearts (*Sugden et al., 2008*). To test whether GSK3β activity is reduced in our cardiac hypertrophy model, we immunoprecipitated GSK3β from the heart lysates and assessed in vitro activity of GSK3β against the peptide of glycogen synthase (GS). Our results revealed that activity of GSK3β was significantly low in ISO-infused heart samples (*Figure 1D*). Moreover, the levels of β-catenin, a GSK3 target transcription factor, which degrade after phosphorylation by GSK3 (*Liu et al., 2002*), was high in ISO-infused hearts (*Figure 1E*). These findings further confirm reduced activity of GSK3β in ISO-infused heart samples. Next, we immunoprecipitated GSK3β from heart lysates and assessed the status of its acetylation and levels of phosphorylation by western blotting. Interestingly, we found markedly increased

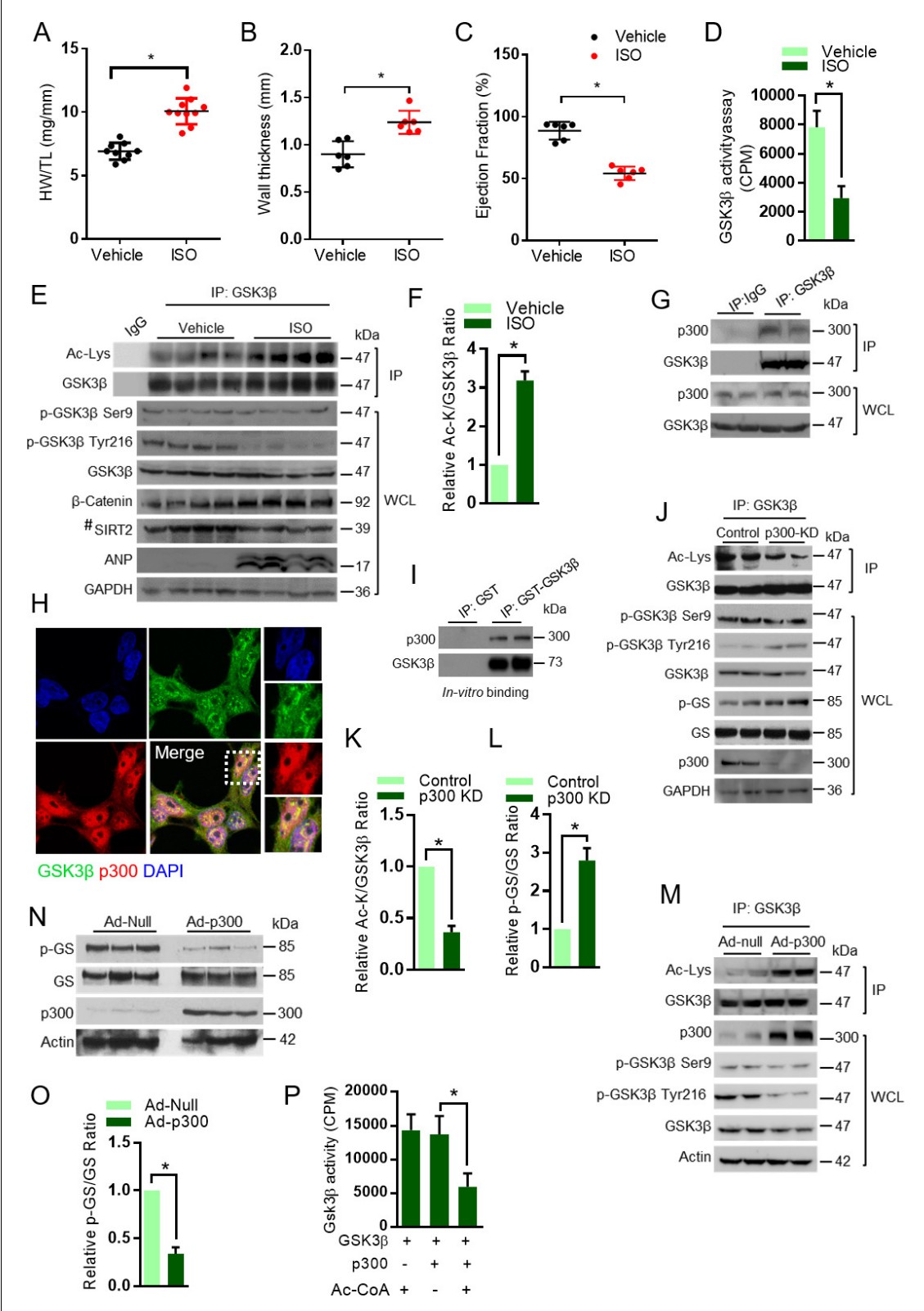

**Figure 1.** Acetylation of GSK3β increased in pathological cardiac hypertrophy. (**A**) Scatter plot showing cardiac hypertrophy, as measured by Heart weight/Tibia Length (HW/TL) ratio of 8 weeks old 129/Sv mice treated with either vehicle or isoproterenol (ISO) at the dose of 10 mg/kg/day. ISO was continuously infused for 7 days using osmotic mini-pumps. $n$ = 9–10 mice per group. Data is presented as mean ± s.d, *p<0.05. Student's $t$ test was used to calculate the p values. (**B**) Scatter plot representing left ventricular posterior wall thickness of 8 weeks old 129/Sv mice treated with either

*Figure 1 continued on next page*

*Figure 1 continued*

vehicle or ISO at the dose of 10 mg/kg/day. ISO was continuously infused for 7 days using osmotic mini-pumps. n = 6 mice per group. Data is presented as mean ± s.d, *p<0.05. Student's *t* test was used to calculate the p values. (C) Scatter plot indicating the contractile functions of heart as represented by ejection fraction of 8 weeks old 129/Sv mice treated with either vehicle or ISO at the dose of 10 mg/kg/day. ISO was continuously infused for 7 days using osmotic mini-pumps. n = 6 mice per group. Data is presented as mean ± s.d, *p<0.05. Student's *t* test was used to calculate the p values. (D) Histogram showing GSK3β activity assay in heart lysates of vehicle or ISO-treated 8 weeks old 129/Sv mice. Mice were treated with either vehicle or ISO at the dose of 10 mg/kg/day for 7 days using osmotic mini-pumps. GSK3β was immunoprecipitated from the heart lysates of vehicle or ISO infused mice using anti-GSK3β antibody, clone GSK-4B (Sigma). The immunoprecipitated GSK3β was incubated with the peptide substrate in the presence of $\gamma-^{32}$P-ATP. The incorporation of $^{32}$P into the GSK3β peptide substrate, which contains specific phosphorylation residues of GSK3β was measured. n = 10 mice per group. Data is presented as mean ± s.d, *p<0.05. Student's *t* test was used to calculate the p values. (E) Eight weeks old 129/Sv mice were treated with either vehicle or ISO at the dose of 10 mg/kg/day for 7 days using osmotic mini-pumps. GSK3β was immunoprecipitated from the heart lysates of vehicle or ISO infused mice using anti-GSK3β antibody (sc-9166, Santa Cruz Biotechnolgy) and the affinity resin immobilized with protein A/G. Western blotting analysis was performed to detect the levels of GSK3β acetylation (Ac-Lys) by anti-acetyl-lysine antibody. IgG was used as negative control in this assay. Heart tissue lysates (WCL) were probed for indicated proteins by western blotting. ANP was used as a positive control to assess cardiac hypertrophy in ISO infused mice. n = 4 mice per group. # marked western blotting images denotes SIRT2 antibody (#12650; Cell Signaling), used in this assay detects single band. (F) Histogram showing relative acetylated GSK3β in vehicle and ISO-treated mice heart tissues, as measured from *Figure 1E*. Signal intensities of acetylated GSK3β and GSK3β were measured by densitometry analysis (ImageJ software). n = 4 mice per group. Data is presented as mean ± s.d. *p<0.05. Student's *t* test was used to calculate the p values. (G) GSK3β was immunoprecipitated from heart tissues of 8 weeks old 129/Sv mice using anti-GSK3β antibody (sc-9166, Santa Cruz Biotechnology), and the affinity resin with protein A/G immobilized. Western blotting was performed to detect GSK3β interaction with p300 using anti-p300 antibody. IgG was used as a negative control. Whole cell lysates (WCL) were probed for the presence of GSK3β and p300 by western blotting. (H) Co-localization of GSK3β with p300 was assessed in 293 T cells by confocal microscopy. The antibodies used are anti-GSK3β (sc-9166, Santacruz), and p300 (05–257, Millipore). DAPI was used to stain the nucleus. Expanded images (right small boxes) show yellow color in the merge image, indicating the co-localization of GSK3β (Green) and p300 (Red) in the nucleus. (I) In vitro binding assay to test the direct interaction between GSK3β and p300. Recombinant p300 (Millipore # 2273152) was incubated with recombinant GST or GST-GSK3β, purified from *E. coli* BL21 (DE3) by affinity chromatography using Glutathione Sepharose 4B. (J) Western blotting analysis showing the acetylation and activity of GSK3β in rat neonatal cardiomyocytes infected with adenovirus expressing either luciferase shRNA (control) or p300 shRNA (p300-KD) for 72 hr. Depletion of p300 was confirmed by western blotting. GSK3β was immunoprecipitated from control and p300-KD cells using anti-GSK3β antibody (sc-9166, Santa Cruz Biotechnology) and the affinity resin immobilized with protein A/G. Western blotting was performed to detect acetylation of GSK3β using the anti Ac-Lysine antibody. GSK3β activity was measured by assessing the phosphorylation of glycogen synthase (p–GS). Site-specific antibodies were used to detect the phosphorylation of GSK3β at indicated residues in cardiomyocyte lysates (WCL). (K) Histogram showing the quantification of relative acetylated GSK3β in control and p300 depleted (p300-KD) rat neonatal cardiomyocytes, as measured from *Figure 1J*. Rat neonatal cardiomyocytes were infected with adenovirus expressing either luciferase shRNA (control) or p300 shRNA (p300-KD) for 72 hr. Signal intensities of acetylated GSK3β and GSK3β were quantified by densitometry analysis (ImageJ software). n = 3 independent experiments. Data is presented as mean ± s.d. *p<0.05. Student's *t* test was used to calculate the p values. (L) Histogram depicting the activity of GSK3β in control and p300 depleted (p300-KD) rat neonatal cardiomyocytes, as measured by the ratio of phosphorylation of glycogen synthase vs total glycogen synthase from *Figure 1J*. Rat neonatal cardiomyocytes were infected with adenovirus expressing either luciferase shRNA (control) or p300 shRNA (p300-KD) for 72 hr. Signal intensities of phospho-glycogen synthase and glycogen synthase were measured by densitometry analysis (ImageJ software). n = 3 independent experiments. Data is presented as mean ± s.d. *p<0.05. Student's *t* test was used to calculate the p values. (M) Western blotting analysis showing the acetylation of GSK3β in rat neonatal cardiomyocytes infected with either control (Ad-null) or p300 overexpressing adenovirus (Ad-p300) for 24 hr. Overexpression of p300 was confirmed by western blotting. GSK3β was immunoprecipitated using anti-GSK3β antibody (sc-9166, Santacruz) and the affinity resin with protein A/G immobilized. Site-specific antibodies were used to detect the phosphorylation of GSK3β at indicated residues in cell lysates (WCL). (N) Western blotting analysis showing the activity of GSK3β in rat neonatal cardiomyocytes infected with control (Ad-null) or p300 expressing adenovirus (Ad-p300) for 24 hr. Overexpression of p300 was confirmed by western blotting and the activity of GSK3β was probed by assessing the levels of p-GS and GS by western blotting. (O) Histogram showing the activity of GSK3β in control (Ad-Null) or p300 overexpressing (Ad-p300) rat neonatal cardiomyocytes, as measured by the ratio of phosphorylation of glycogen synthase vs total glycogen synthase from *Figure 1N*. Signal intensities of phospho-glycogen synthase and glycogen synthase were assessed by densitometry analysis (ImageJ software). n = 3 independent experiments. Data is presented as mean ± s.d. *p<0.05. Student's *t* test was used to calculate the p values. (P) In vitro kinase assay showing the activity of acetylated and non-acetylated GSK3β. Human GSK3β with HA tag was overexpressed in HeLa cells by transfection of the plasmid pcDNA3-HA-GSK3β. HA-GSK3β was immunoprecipitated using HA-coupled agarose beads (Sigma-Aldrich) and the HA-GSK3β was acetylated by recombinant p300 (Millipore), in the presence or absence of Acetyl-CoA (Ac-CoA) in HAT buffer. The enzymatic activity of GSK3β was measured against glycogen synthase (GS)-peptide. n = 6 independent experiments. Data is presented as mean ± s.d. *p<0.05. One-way ANOVA was used to calculate the p values.

DOI: https://doi.org/10.7554/eLife.32952.002

acetylation of GSK3β in the heart samples of ISO-infused mice (*Figure 1E and F*). However, we did not see marked changes in Ser9 phosphorylation of GSK3β in the chronic ISO-infused heart samples (*Figure 1E*). Our results were consistent with previous observations, where the total protein levels and Ser9 phosphorylation of GSK3β were not changed in the heart after pressure overload or

myocardial infarction (*Zhai et al., 2007*). Similarly, the GSK3β Ser9 phosphorylation was unchanged in the heart of mice-treated with ISO (*Zhang et al., 2011*). These findings indicate that the reduced activity of GSK3β found in chronic ISO-infused hearts is correlated with increased lysine acetylation, but not with inhibitory Ser nine phosphorylation. Furthermore, we observed reduced levels of Tyr216 phosphorylation of GSK3β in ISO-infused heart samples. The expression levels of SIRT2 also reduced in the ISO-treated heart samples (*Figure 1E*), suggesting that acetylation of GSK3β may be linked to reduced levels of SIRT2 deacetylase.

Acetyltransferase p300 has been shown to play a major role in the cardiac homeostasis (*Yanazume et al., 2003a*, *2002*). Our immunoprecipitation experiments showed that p300 binds to GSK3β (*Figure 1G*). Further confocal microscopy analysis indicated that p300 co-localizes with GSK3β (*Figure 1H*). Next, we assessed direct binding of p300 with GSK3β using recombinant puri-fied proteins. Our results indicate that p300 binds to GSK3β directly in an in vitro reaction (*Figure 1I*). In the next set of experiments, GSK3β was immunoprecipitated from p300-depleted cells and its acetylation status and activity were analyzed. We found that the depletion of p300 resulted in reduced levels of acetylation, although the effect is not robust, it increased the activity of GSK3β, as suggested by increase in GS phosphorylation (*Figure 1J–1L*). We believe that GSK3β acetylation may be regulated by multiple acetyltransferases including p300. Interestingly, depletion of p300 markedly increased the phosphorylation of Tyr216, but not Ser9 in GSK3β (*Figure 1J*). To validate our findings, we performed p300 overexpression experiments in cardiomyocytes. We found significantly increased acetylation of GSK3β, which is associated with reduced activity of GSK3β in p300 overexpressing cells (*Figure 1M, N and O*). Moreover, overexpression of p300 markedly reduced the Tyr216, but not Ser9 phosphorylation of GSK3β (*Figure 1M*). Next, we tested the enzy-matic activity of acetylated GSK3β in vitro and found that p300-mediated acetylation significantly impairs kinase activity of GSK3β (*Figure 1P*). These findings suggest that p300-mediated acetylation might influence the phosphorylation of GSK3β at Tyr216 residue, but not Ser9, to inhibit the enzy-matic activity.

## Acetylation of GSK3β influences ATP binding: Insights from molecular modeling and molecular dynamics simulation of GSK3β WT and acetylated K183 mutant

In our previous work, we found that acetylation of GSK3β at K15 promotes mitochondrial localization and is regulated inside the mitochondria by SIRT3 deacetylase. SIRT3-mediated deacetylation of GSK3β regulates the development of organ fibrosis (*Sundaresan et al., 2016*). Since SIRT2 is a cyto-plasmic protein, we suspected that SIRT2 might not be regulting K15 acetylation of GSK3β. There-fore, we performed tandem mass spectrometry (MS/MS) analysis of GSK3β and identified two lysine residues, K150 and K183, as acetylation sites in GSK3β (*Figure 2A*). While K183 is located in the vicinity of nucleotide-binding pocket of GSK3β, K150 is located on the surface away from both nucle-otide-binding pocket as well as substrate-binding region (*Figure 2B* I-III). Magnesium is an important cofactor that helps in neutralizing the negative charges of the phosphate residues of nucleotide. Interestingly, the side-chain of K183 residue is proximal to the magnesium ion of the adenosine di-phosphate nucleotide (*Figure 2B and C*). The addition of the acetylated group to the NZ atom of K183 side-chain may result in destabilization of $Mg^{2+}$ from ADP (*Figure 2B* II-IV). Overall, in-silico analysis suggests that the acetylation of K183 side-chain may reduce the affinity of adenosine tri-phosphate (ATP) nucleotide to the binding pocket by interfering with its magnesium cofactor.

To better understand the effect of acetylation of K183 on the binding of adenine nucleotide in the pocket of GSK3β, we performed molecular dynamics simulations for the ADP-loaded wild type and acetylated K183 mutant (acK183). Initial model for the wild type was generated from the crystal structure of GSK3β (PDB ID 4NMO). For the acK183 mutant, the wild-type model was used as a tem-plate to generate in silico acetylation on the K183 residue (further details in Materials and methods). We ran five independent trajectories of 20 ns each totaling to 100 ns for each system. Multiple pro-duction runs would provide greater sampling of conformations for each system and further present an opportunity to capture the stochastic events, which otherwise might not be captured in a single long trajectory. Visual analyses of the molecular dynamics (MD) trajectories show drastic fluctuations in the ADP nucleotide with adenosine base completely destabilized in the acK183 mutant (*Figure 2D*). To quantify these effects, we first performed the root mean square deviation (RMSD) measurements for the Cα backbone atoms of the protein and ADP nucleotide, in both the systems.

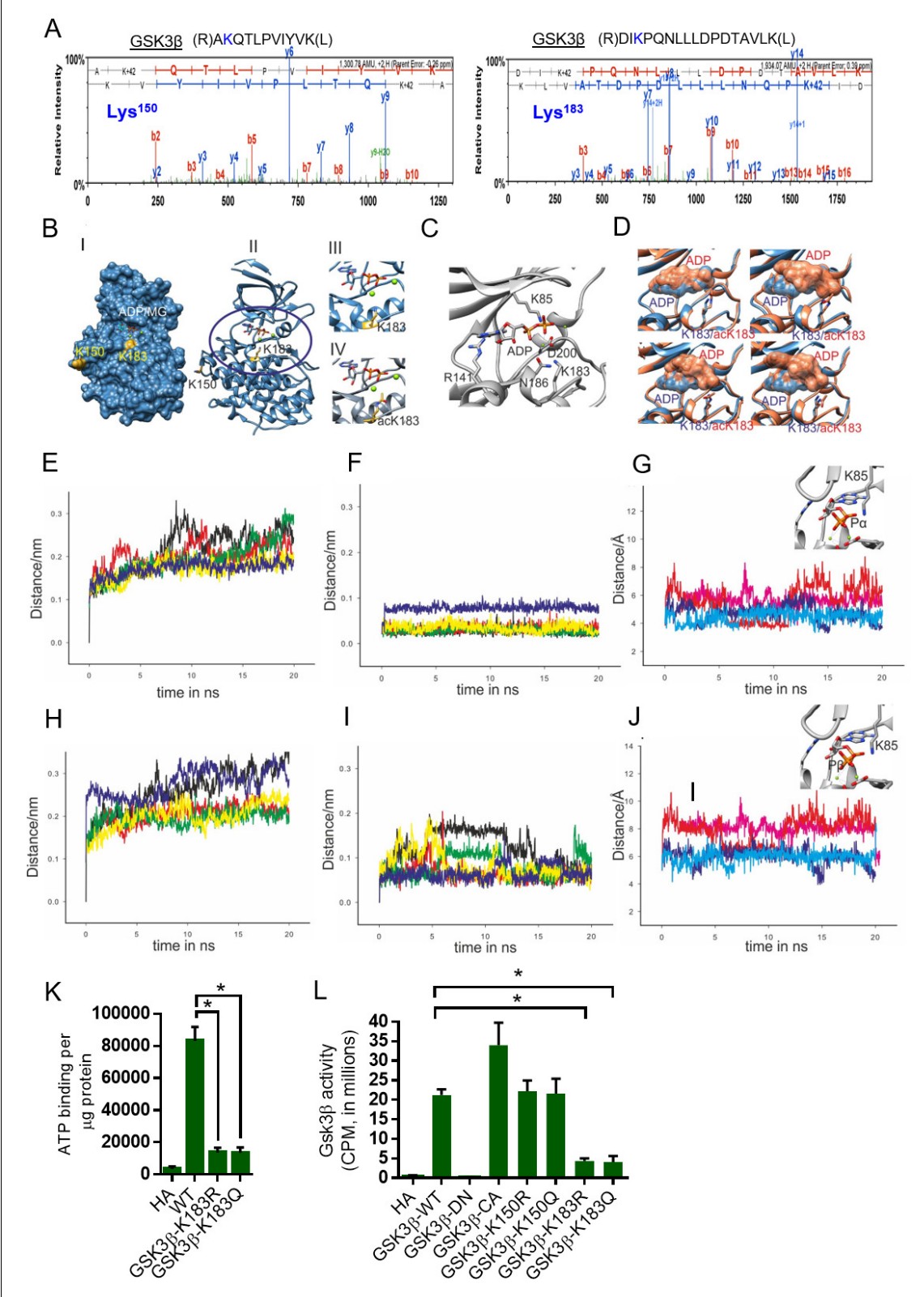

**Figure 2.** Molecular modeling and molecular dynamics simulations of GSK3β wild-type and K183 acetylated mutant. (A) Annotation of representative tandem mass spectra of trypsin-digested GSK3β, depicting K150 and K183 acetylation. (B) Representation of the acetylation sites on the crystal structure of GSK3β (PDB ID 4NM0): (i) surface (ii) cartoon representation. (iii) Magnified active site representing position of K183 and (iv) magnified active site representing position of acetylated K183 (acK183). (C) Nucleotide-binding site in GSK3β crystal structure (PDB ID 4NM0) representing ADP,

*Figure 2 continued on next page*

Figure 2 continued

nucleotide interacting residues and K183. (D) Overlay of the wild-type (blue) and acK183 mutant (orange) of GSK3β representing the surface of ADP nucleotide, at random snapshots in the MD trajectory. (E) Overlay of protein backbone Cα RMSD plots of the five 20 ns MD trajectories in wild type. (F) Overlay of ADP nucleotide RMSD plots of the five 20 ns MD trajectories in wild type. (G) Overlay of the distance between the NZ atom of K85 and α-phosphate of ADP as a function of time for two stable trajectories (dark blue/cyan – wild type, pink/red – acK183). (H) Overlay of protein backbone Cα RMSD plots of the five 20 ns MD trajectories in acK183 mutant. (I) Overlay of ADP nucleotide RMSD plots of the five 20 ns MD trajectories in acK183 mutant. (J) Overlay of the distance between the NZ atom of K85 and β-phosphate of ADP as a function of time for two stable trajectories (dark blue/cyan – wild type, pink/red – acK183). (K) Histogram showing binding of γ−$^{32}$P-ATP to recombinant wild type and mutants of His-GSK3β. Plasmids encoding wild type and mutants of His-GSK3β were transformed into *E. coli* BL21 (DE3). His-GSK3β and its mutants were purified by Ni-NTA affinity chromatography. $n$ = 4 independent experiments. Data is presented as mean ± s.d. *p<0.05. One-way ANOVA was used to calculate the p values. (L) Histogram showing activity of HA-tagged WT or mutants of GSK3β. HA-tagged human GSK3β or its mutants were overexpressed in HeLa cells by transfection of their respective plasmids. HA-GSK3β or its mutants were immunoprecipitated using HA-coupled agarose beads (Sigma-Aldrich). The enzymatic activity of GSK3β was measured against glycogen synthase (GS)-peptide as described in the Materials and methods section. GSK3β-DN - GSK3β-K85A; Dominant negative. GSK3β-CA- GSK3β S9A; catalytically active. $n$ = 4 independent experiments. Data is presented as mean ± s.d. *p<0.05. One-way ANOVA was used to calculate the p values.

DOI: https://doi.org/10.7554/eLife.32952.003

The following figure supplements are available for figure 2:

**Figure supplement 1.** Protein backbone Cα RMSF plots of the wild type (dark blue) and Ac-K183 mutant (red).

DOI: https://doi.org/10.7554/eLife.32952.004

**Figure supplement 2.** Homology alignment of GSK3β between different species.

DOI: https://doi.org/10.7554/eLife.32952.005

**Figure supplement 3.** Stoichiometry for GSK3β-K150, -K183 acetylation.

DOI: https://doi.org/10.7554/eLife.32952.006

RMSD plots of the Cα backbone atoms for the five individual trajectories of 20 ns were plotted separately for the wild type and acK183 mutant (*Figure 2E and H*). The RMSD deviations were similar for both the plots ranging between 0.1 and 0.35 nm (*Figure 2F and I*). In agreement, we found only moderate changes in the global root mean square fluctuations (RMSF) of all the backbone Cα atoms of the acK183 mutant (*Figure 2—figure supplement 1*). Interestingly, we observed significant variations in the RMSD plots of the ADP nucleotide in the acK183 mutant compared to the wild type (*Figure 2F and I*). The drastic deviations observed in the RMSD plots of ADP nucleotide in acK183 mutant concurs with the fluctuations seen in the snap-shots of structures during MD simulations (*Figure 2D*). For further analysis, we considered two trajectories with stable Cα backbone RMSD's for each of the wild type (yellow, blue) (*Figure 2E*) and acK183 mutant (red, green) of GSK3β (*Figure 2E*). We plotted time-dependent distance measurements between protein-nucleotide interactions in the binding pocket of GSK3β. We found that acetylation causes increase in distance between the side-chain amine of K85 and the phosphates of ADP nucleotide (*Figure 2G and J*). Positively charged K85 residue is essential for neutralizing the negative charge of phosphates of nucleotides. The escalation in distances between ADP and K85 suggest reduced stability for negatively charged phosphates in the acetylated mutant. Our results suggest that the acetylation would hinder both the adenosine binding as well as prevent stable interactions of the negatively charged phosphates with the protein. Collectively, we propose that the acetylation of K183 would destabilize the binding of the adenine nucleotide to the pocket of GSK3β.

To test this possibility further, we performed ATP binding assay with wild type and mutant versions of GSK3β, where K183 was mutated either to glutamine (Q, acetylation mimetic) or to arginine (R, deacetylation mimetic). GSK3β-K183 mutations significantly reduced binding of ATP to GSK3β, when compared to wild type GSK3β (*Figure 2K*). These findings provided strong evidence that K183 is critically important for ATP binding to GSK3β and any modifications, including acetylation might affect its ATP-binding ability. To test whether K183 acetylation affects the catalytic activity, we performed in vitro kinase assay with mutants of GSK3β. In this assay, kinase-dead GSK3β-K85A was used as a negative control and GSK3β-S9A was used as the catalytically active mutant. Our results indicate that mutation of K183 to either arginine or glutamine drastically reduced the catalytic activity toward a glycogen synthase peptide (*Figure 2L*). Mutation of K150, which is less conserved among species (*Figure 2—figure supplement 2*) and exhibited low stoichiometry of acetylation (*Figure 2—figure supplement 3*), had no effect on GSK3β activity (*Figure 2L*). It is interesting to note that mutation of K183 to either K183R or K183Q, reduced the catalytic activity more than four folds,

indicating that K183 is critical for the functionality of GSK3β. Collectively, these findings suggest that K183 is an important residue for ATP binding and the catalytic activity of GSK3β.

## SIRT2 modulates the kinase activity of GSK3β by reversible acetylation

In our work, we found reduced levels of SIRT2 deacetylase in the ISO-treated mice hearts. Since SIRT2 and GSK3β share cytoplasmic localization, we suspected that SIRT2 may be the GSK3β deacetylase. To test our hypothesis, we overexpressed all the sirtuin isoforms and assessed the acetylation, phosphorylation and activity of GSK3β. Our results suggest that SIRT2 overexpression markedly reduced the acetylation of GSK3β, while increasing its activity against glycogen synthase (*Figure 3—figure supplement 1*). Interestingly, we do not observe any changes in the phosphorylation of Ser9 residue of GSK3β. However, we found increased phosphorylation of GSK3β at Tyr216. Consistent with our previous work (*Sundaresan et al., 2016*), we found SIRT3 to be capable of deacetylating GSK3β and enhancing its catalytic activity (*Figure 3—figure supplement 1*). To further verify our findings, we immunoprecipitated GSK3β from SIRT2-deficient heart lysates and tested the levels of acetylation. As expected, we found markedly increased acetylation of GSK3β in SIRT2-KO hearts (*Figure 3A and B*). To test whether SIRT2 interacts with GSK3β, we immunoprecipitated GSK3β from heart samples and tested its binding with SIRT2 by western blotting. We found that SIRT2 interacts with GSK3β (*Figure 3C*). We also found that recombinant purified GSK3β is capable of binding to SIRT2 in vitro, suggesting direct interaction between both proteins (*Figure 3D*). To test whether SIRT2 deacetylates GSK3β, we performed an in vitro deacetylation assay, where the acetylated GSK3β was incubated with either WT or SIRT2-H187Y, catalytic mutant of SIRT2 with and without NAD$^+$. Western blotting analysis indicated WT, but not catalytic mutant-SIRT2 markedly reduced the acetylation status of GSK3β in an NAD$^+$-dependent manner (*Figure 3E*). These findings indicate that SIRT2 is a GSK3β deacetylase. Next, we tested whether SIRT2-mediated deacetylation enhances the catalytic activity of GSK3β by an in vitro kinase assay. The acetylated and deacetylated GSK3β were incubated with GS peptide and the GSK3β activity was monitored. Results indicated that acetylation significantly reduced the activity of GSK3β. However, incubation of acetylated-GSK3β with wild type, but not SIRT2-H187Y catalytic inactive mutant of SIRT2 restored the activity of GSK3β in a NAD$^+$-dependent manner (*Figure 3F*), suggesting that SIRT2-mediated deacetylation increases GSK3β activity.

To further test the activity of GSK3β, we measured the levels of phosphorylation of glycogen synthase in SIRT2-depleted cardiomyocytes. SIRT2 depletion increases the cellular global lysine acetylation and the acetylation of GSK3β (*Figure 3G*). Moreover, GSK3β-specific phosphorylation of glycogen synthase and β-catenin was reduced in SIRT2-depleted cardiomyocytes, which is associated with reduced Tyr216, but not Ser9 phosphorylation of GSK3β (*Figure 3G*). To further validate the results in vivo, we tested the phosphorylation of glycogen synthase in SIRT2-deficient heart samples, and observed reduced phosphorylation of glycogen synthase, which is associated with reduced Tyr216, but not Ser9 phosphorylation of GSK3β in SIRT2-KO hearts (*Figure 3H*). To further confirm the reduced activity of GSK3β in SIRT2-deficient hearts, we performed an *in vitro* kinase assay. Consistent with our findings, we observed significantly reduced activity of GSK3β toward a peptide of glycogen synthase (*Figure 3I*). Next, we measured the total and phosphorylated protein levels of β-catenin, a GSK3 target transcription factor, which is degraded, after phosphorylation by GSK3 (*Liu et al., 2002*), in the heart samples of SIRT2-deficient mice. Our results suggest that the protein levels of β-catenin are high in SIRT2-KO heart lysates, while the GSK3 target phosphorylation of β-catenin was decreased drastically (*Figure 3H*), indicating that SIRT2-mediated deacetylation might control the kinase activity of GSK3β.

To further understand the mechanism involved in acetylation-mediated regulation of GSK3β activity, we examined the subcellular localization of GSK3β. Treatment of cells with a SIRT2 inhibitor (AGK2) enhanced the acetylation of tubulin, a well-accepted target of SIRT2, and the acetylation of GSK3β, while decreasing its activity against glycogen synthase (*Figure 3—figure supplement 2*). Consistent with previous results, we did not observe any changes in the phosphorylation of Ser9 residue of GSK3β following AGK2 treatment. However, we found decreased phosphorylation of GSK3β at Tyr216 (*Figure 3—figure supplement 2*). Interestingly, AGK2 treatment had no effect on subcellular distribution of the GSK3β (*Figure 3—figure supplement 3*). Next, we tested the localization of K183 mutants of GSK3β. As expected, the results suggest that the WT and K183 mutants of GSK3β exhibited cytoplasmic localization (*Figure 3—figure supplement 4*). These results indicate that

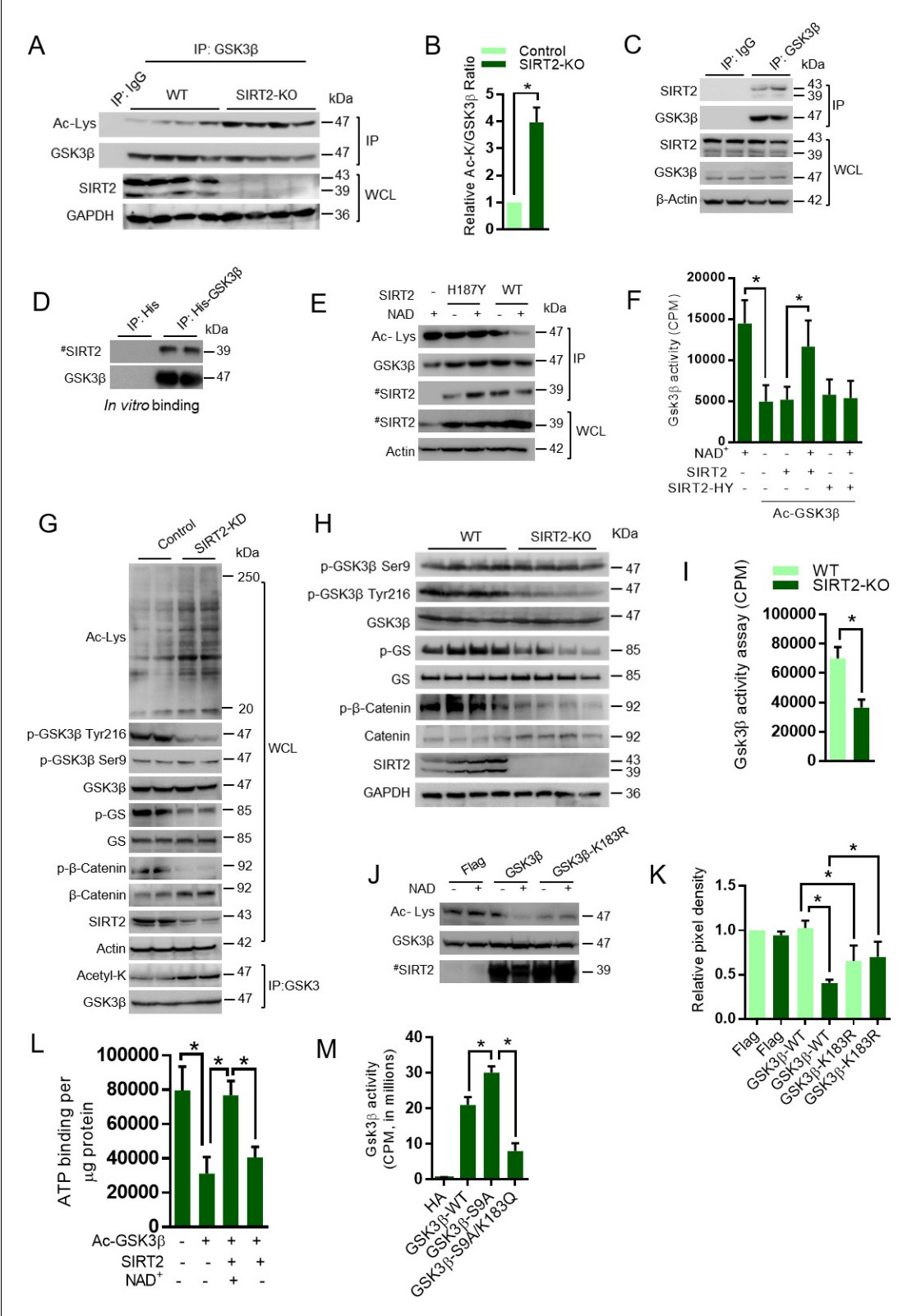

**Figure 3.** SIRT2 binds to, deacetylates and activates GSK3β. (**A**) Western blot analysis of acetylated GSK3β in heart samples of 9 months old WT and SIRT2-KO littermates. GSK3β was immunoprecipitated from heart tissue lysates of WT and SIRT2-KO mice using anti-GSK3β antibody (sc-9166, Santa Cruz Biotecholnlgy), and the affinity resin immobilized with protein A/G. Western blotting was performed to detect GSK3β acetylation by anti-Ac-Lysine antibody. IgG was used as a negative control. Whole cell lysates (WCL) were probed for the SIRT2 and GAPDH by western blotting. *n* = 4 mice per
*Figure 3 continued on next page*

*Figure 3 continued*

group. (B) Histogram showing relative acetylated GSK3β in 9 months old WT and SIRT2-KO mice heart tissues, as measured from *Figure 3A*. Signal intensities of acetylated GSK3β and GSK3β were measured by densitometry analysis (ImageJ software). $n = 4$ mice per group. Data is presented as mean ± s.d, *$p<0.05$. Student's $t$ test was used to calculate the p values. (C) GSK3β was immunoprecipitated from heart tissue lysates of 8 weeks old 129/Sv mice using anti-GSK3β antibody (sc-9166, Santa Cruz Biotechnolgy ), and the affinity resin immobilized with protein A/G. GSK3β interaction with SIRT2 was tested by western blotting using anti-SIRT2 antibody. IgG was used as negative control. Heart lysates was probed for indicated proteins by western blotting. (D) In vitro binding assay to test the interaction between GSK3β and SIRT2. Flag-SIRT2 was overexpressed in 293 cells by a plasmid encoding human Flag-SIRT2. Recombinant His or His-GSK3β was purified from *E. coli* BL21 (DE3) by Ni-NTA affinity chromatography and were incubated with 293 T cell lysates overexpressing human Flag-SIRT2. Interaction between GSK3β and SIRT2 was tested by western blotting. # marked western images denotes SIRT2 antibody used in this assay detects single band. (E) In vitro deacetylation assay showing SIRT2 as GSK3β deacetylase. Human HA-GSK3β was overexpressed in HeLa cells by transfection of the plasmid pcDNA3-HA-GSK3β. HA-GSK3β was immunoprecipitated using HA-coupled agarose beads (Sigma-Aldrich) and the HA-GSK3β was acetylated by recombinant p300 (Millipore), in the presence or absence of Acetyl-CoA (Ac-CoA) in HAT buffer. The acetylated HA-GSK3β was further incubated with either Flag-tagged SIRT2 or SIRT2-H187Y, which were immunoprecipitated from HEK 293 cell lysates overexpressing respective plasmids encoding Flag-tagged WT or SIRT2-H187Y using agarose beads conjugated to anti-Flag antibody (Sigma A2220). The deacetylation reaction was carried out in the presence or absence of NAD$^+$ in a HDAC buffer. GSK3β acetylation was analyzed by western blotting using anti-Ac-Lysine antibody. # marked western images denotes SIRT2 antibody used in this assay detects single band. (F) In vitro kinase assay depicting the activity of acetylated and deacetylated GSK3β. Human HA-GSK3β was overexpressed in HeLa cells by transfection of the plasmid pcDNA3-HA-GSK3β. Recombinant HA-GSK3β was immunoprecipitated using HA-coupled beads and was acetylated by recombinant p300 in the presence or absence of Acetyl-CoA (Ac-CoA) in HAT buffer. Acetylated GSK3β was further deacetylated by either Flag-tagged WT or SIRT2-H187Y (SIRT2-HY), a catalytic inactive mutant of SIRT2, which was immunoprecipitated from HEK 293 cells, overexpressed with plasmid encoding Flag-tagged WT or SIRT2-H187Y using agarose beads conjugated to anti-Flag antibody (Sigma A2220). The deacetylation reaction was carried out in the presence or absence of NAD$^+$ in a HDAC buffer and further enzymatic activity of GSK3β was measured against glycogen synthase (GS)-peptide, as described in the Materials and methods section. $n = 5$. Data is presented as mean ± s.d. *$p<0.05$. One-way ANOVA was used to calculate the p values. (G) Western blot analysis of acetylated GSK3β from control or SIRT2-depleted (SIRT2-KD) cardiomyocytes. Neonatal rat cardiomyocytes were transfected with either non-targeting (control) or siRNA targeting SIRT2 using Lipofectamine RNAiMAX reagent for 72 hr. SIRT2 depletion was confirmed by Western blotting. Total cellular acetylation was probed by anti-Ac-Lysine antibody to test the effect of SIRT2 depletion in cardiomyocytes. GSK3β was immunoprecipitated from these cell lysates using anti-GSK3β antibody (sc-9166, Santa Cruz Biotechnolgy), and the affinity resin immobilized with protein A/G. Western blotting was performed to detect acetylation of GSK3β by anti-Ac-Lysine antibody. Cell lysates (WCL) from control and SIRT2-KD cardiomyocytes were probed for indicated proteins by western blotting. (H) Western blotting analysis of hearts lysates from 9 months old WT and SIRT2-KO mice littermates for indicated proteins. $n = 4$ mice per group. (I) Histogram showing activity of GSK3β in WT and SIRT2-KO mice hearts at 9 months of age. GSK3β was immunoprecipitated from the heart lysates of WT and SIRT2-KO mice using anti-GSK3β antibody, clone GSK-4B (Sigma). The immunoprecipitated GSK3β was incubated with the peptide substrate in the presence of γ−$^{32}$P-ATP. The incorporation of $^{32}$P into the GSK3β Peptide Substrate, which contains specific phosphorylation residue of GSK3β was measured. $n = 6$ mice per group. Data is presented as mean ± s.d. *$p<0.05$. Student's $t$ test was used to calculate the p values. (J) In vitro deacetylation assay to test whether SIRT2 deacetylates K183 residue of GSK3β. HA-tagged GSK3β or GSK3β-K183R was overexpressed in HeLa cells and was immunoprecipitated using HA-coupled beads. HA-tagged WT-GSK3β or GSK3β-K183R were incubated with Flag-SIRT2 immunoprecipitated from HEK 293 T cells using agarose beads conjugated to Anti-Flag antibody (Sigma A2220). The deacetylation reaction was carried out in the presence or absence of NAD$^+$ in a deacetylation buffer. Acetylation status of GSK3β was analyzed by western blotting. # marked western images denotes SIRT2 antibody used in this assay detects single band. (K) Histogram showing relative acetylation of HA-tagged GSK3β or GSK3β-K183R, which was incubated with Flag-SIRT2. The data is generated from *Figure 3J*. Signal intensities of acetylated-GSK3β and GSK3β were measured by densitometry analysis (ImageJ software). $n = 4$ independent experiments. Data is presented as mean ± s.d. *$p<0.05$. One-way ANOVA was used to calculate the p values. (L) Histogram showing binding of γ−$^{32}$P-ATP to acetylated and deacetylated His-GSK3β. Recombinant His-GSK3β was purified from *E. coli* BL 21 (DE3) by Ni-NTA affinity chromatography. Purified His-GSK3β was acetylated by recombinant p300 in the presence of Ac-CoA in HAT buffer. Acetylated His-GSK3β was further deacetylated by Flag-SIRT2 immunoprecipitated from HEK 293 T cells. The binding of γ−$^{32}$P-ATP to acetylated and deacetylated His-GSK3β was assessed by the protocol described in Materials and methods section. $n = 4$. Data is presented as mean ± s.d. *$p<0.05$. One-way ANOVA was used to calculate the p values. (M) Histogram showing activity of WT or mutants of GSK3β. HA-tagged WT or mutants of GSK3β was immunoprecipitated from HeLa cells transfected with respective plasmids using HA-coupled agarose beads. The enzymatic activity of GSK3β was measured against glycogen synthase (GS)-peptide, as described in the Materials and methods section. $n = 4$. Data is presented as mean ± s.d. *$p<0.05$. One-way ANOVA was used to calculate the p values.

DOI: https://doi.org/10.7554/eLife.32952.007

The following figure supplements are available for figure 3:

**Figure supplement 1.** Western blotting analysis showing acetylation and phosphorylation of GSK3β and its downstream target GS in HeLa cells overexpressing the Sirtuin isoforms, SIRT1-SIRT7.
DOI: https://doi.org/10.7554/eLife.32952.008

**Figure supplement 2.** Western blotting analysis of vehicle or SIRT2 inhibitor, AGK2-(10 μM, 12 hr) treated rat neonatal cardiomyocytes for indicated proteins.
DOI: https://doi.org/10.7554/eLife.32952.009

**Figure supplement 3.** Representative confocal images of vehicle or AGK2 (10 μM, 12 hr) treated HeLa cells stained with GSK3β (Green).
DOI: https://doi.org/10.7554/eLife.32952.010

*Figure 3 continued on next page*

*Figure 3 continued*

**Figure supplement 4.** Representative confocal images of HA-tagged WT or mutants of GSK3β transiently overexpressed in GSK3β-deficient mouse embryonic fibroblasts.
DOI: https://doi.org/10.7554/eLife.32952.011
**Figure supplement 5.** Western blot analysis of 9 months old WT and SIRT2-KO mice heart samples for indicated proteins.
DOI: https://doi.org/10.7554/eLife.32952.012

change in the activity of GSK3β is not associated with its localization. To test whether SIRT2 deficiency is compensated by other sirtuin isoforms, we checked for the levels of all the sirtuins in SIRT2-deficient mice hearts. However, we found no difference in expression levels of any other sirtuin isoforms in SIRT2-deficient mice heart samples (*Figure 3—figure supplement 5*).

Our MS/MS analysis indicated that GSK3β K150 and K183 residues are acetylated and GSK3β K183 is involved in the regulation of its kinase activity. To test whether SIRT2 deacetylates K183, we performed an in vitro assay, where wild type and GSK3β K183R mutants were incubated with SIRT2. Our results revealed that GSK3β-K183R showed basal level acetylation, which is appreciably lower than the acetylation of wild type GSK3β. These findings confirm our observations that GSK3β is acetylated at multiple residues endogenously, and K183 residue is one among them. SIRT2 deacetylates the wild type GSK3β, but not GSK3β-K183R in the presence of NAD$^+$, indicating that K183R may be the key target residue for SIRT2 deacetylase (*Figure 3J and K*). Collectively, these findings suggest that catalytic activity of GSK3β is governed by acetylation at GSK3β K183 residue. To test whether SIRT2-mediated deacetylation enhances the ATP binding to GSK3β, we analyzed the ATP-binding ability of acetylated and deacetylated GSK3β. As expected, we found that acetylation inhibits the ATP binding, whereas SIRT2-mediated deacetylation restores the ATP-binding ability of GSK3β (*Figure 3L*).

GSK3β kinase activity is known to be inhibited by N-terminal Ser9 phosphorylation (*Ali et al., 2001*; *Song et al., 2015*). Therefore, we tested the phosphorylation of GSK3β in SIRT2-deficient heart samples with an antibody that recognizes the phosphorylation at Ser9. Surprisingly, we did not observe any differences in the phosphorylation at Ser9 of GSK3β in control and SIRT2-deficient cardiomyocytes or hearts (*Figure 3G and H*), suggesting that the reduced activity of GSK3β is associated with acetylation, but not the phosphorylation at Ser9 residue. We next tested whether the activity of constitutively active GSK3β S9A mutant is inhibited by acetylation mimetic mutation (GSK3β-S9A/K183Q). Our results suggest that the activity of GSK3β-S9A mutant, which is not inhibited by the Ser9 phosphorylation, was reduced almost three-folds when K183 was mutated to K183Q (*Figure 3M*). These findings suggest that K183 residue is critical for ATP binding in constitutively active GSK3β S9A mutant. We presume that acetylation might inhibit the activity of constitutively active GSK3β S9A mutant independent of inhibitory phosphorylation, but further experiments are required to prove this assumption.

Next, we tested the phosphorylation of GSK3β at Tyr216 residue, which is known to enhance GSK3 activity by promoting substrate accessibility (*Dajani et al., 2001*; *Hughes et al., 1993*). Surprisingly, we found markedly reduced phosphorylation of GSK3β at Tyr216 residue in the cardiomyocytes or heart lysates of SIRT2-deficient mice (*Figure 3G and H*). These results suggest that acetylation might have negative impact on the activating tyrosine phosphorylation of GSK3β.

## Molecular modeling of GSK3α suggests that the acetylation of residue K246 would have similar consequences as K183 of GSK3β

GSK3 has two isoforms, α and β, which have overlapping cellular substrates and functions (*Rayasam et al., 2009*). In the heart, both α and β isoforms of GSK3 are expressed (*Sugden et al., 2008*). To test whether both isoforms of GSK3 are acetylated in SIRT2-depleted cells, we immunoprecipitated GSK3 isoforms and acetylation status was tested by western blotting. We found both GSK3α and GSK3β to be significantly acetylated in SIRT2-depleted cells as compared to control cells (*Figure 4A and B*). Next, we tested whether SIRT2-mediated deacetylation increases the catalytic activity of GSK3α. The acetylated and deacetylated GSK3α was incubated with GS peptide and the GSK3α activity was measured. Results suggest that acetylation significantly reduced the activity of GSK3α and incubation of acetylated-GSK3α with wild type SIRT2 restored its activity in an NAD$^+$-dependent manner (*Figure 4C*), suggesting that SIRT2-mediated deacetylation enhances the activity

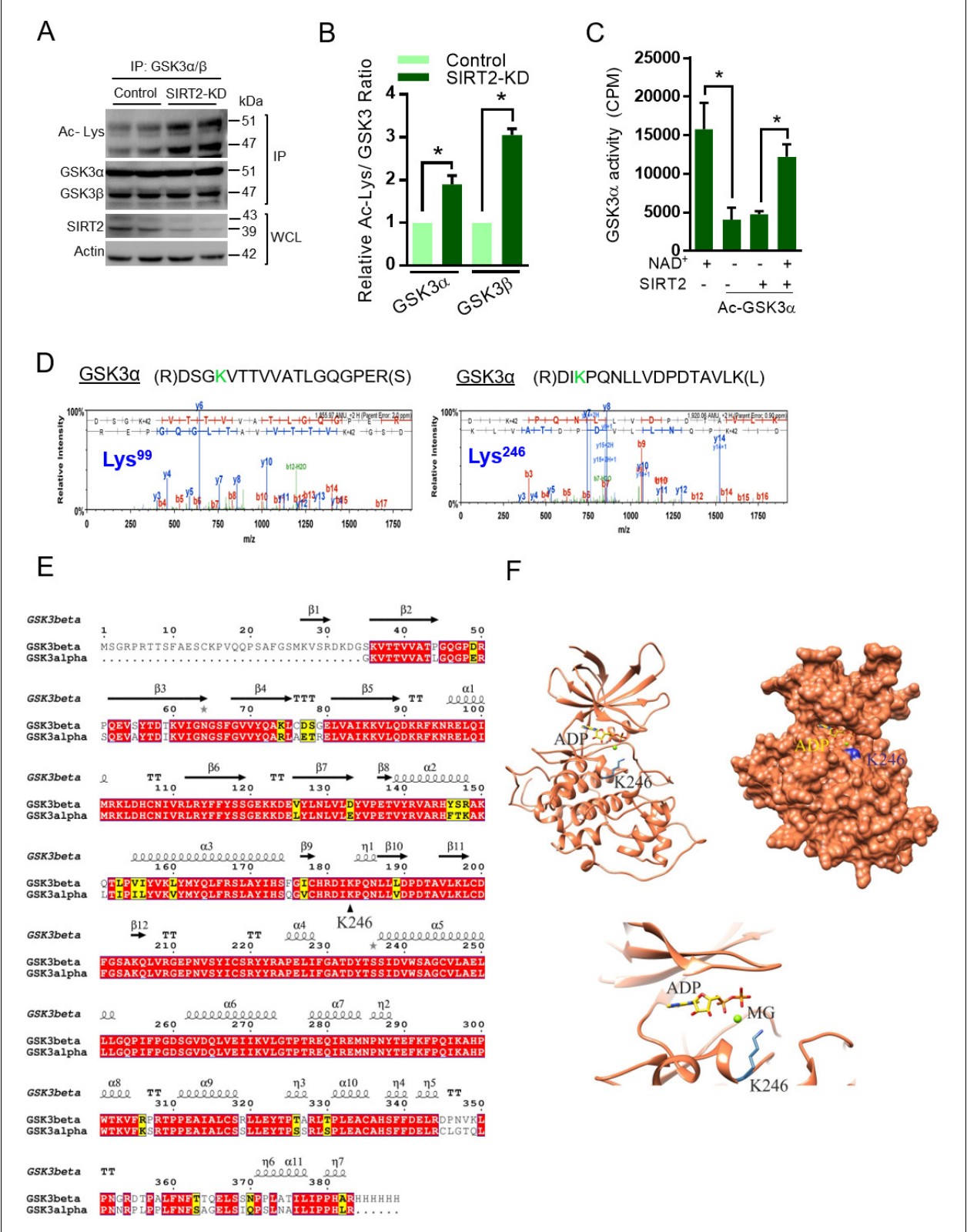

**Figure 4.** Molecular modeling of acetylated GSK3α. (**A**) Western blot analysis showing acetylation status of both isoforms of GSK3 in control or SIRT2 depleted (SIRT2-KD) cardiomyocytes. Neonatal rat cardiomyocytes were transfected with either non-targeting or siRNA pool targeting SIRT2 using Lipofectamine RNAiMAX reagent for 72 hr. SIRT2 depletion was confirmed by western blotting. GSK3 was immunoprecipitated from cell lysates using anti-GSK3 antibody and the affinity resin immobilized with protein A/G. Western blotting was performed to detect GSK3α/β acetylation by anti-Ac-
*Figure 4 continued on next page*

*Figure 4 continued*

Lysine antibody. Cell lysates was probed for SIRT2 and actin antibodies by western blotting. (B) Histogram showing relative acetylated-GSK3α and GSK3β in control and SIRT2-depleted (SIRT2-KD) cardiomyocytes, as measured from *Figure 4A*. Signal intensities of acetylated-GSK3α and acetylated-GSK3β were measured by densitometry analysis (ImageJ software). *n* = 3. Data is presented as mean ± s.d. *p<0.05. Student's *t* test was used to calculate the p values. (C) Histogram showing enzymatic activity of acetylated and deacetylated GSK3α. Recombinant HA- GSK3α was immunoprecipitated from HeLa cells overexpressing pcDNA-HA-GSK3α using HA-coupled agarose beads. Immunoprecipitated HA-GSK3α was acetylated by p300 in the presence of Acetyl-CoA (Ac-CoA) in HAT buffer. Acetylated GSK3α was further deacetylated by Flag-SIRT2 immunoprecipitated from HEK 293 T cells overexpressing plasmid encoding Flag-tagged SIRT2-WT using agarose beads conjugated to anti-Flag antibody (Sigma A2220). The enzymatic activity of GSK3α was measured against glycogen synthase (GS)-peptide. *n* = 5. Data is presented as mean ± s.d. *p<0.05. One-way ANOVA was used to calculate the p values. (D) Annotation of representative tandem mass spectra of trypsin-digested GSK3α, depicting K99, K246 acetylation. (E) Protein sequence alignment of the modeled region of GSK3α and the structure of GSK3β. (F) Cartoon, surface representation of the homology model of GSK3α (highlighted is the adenine nucleotide-binding pocket and position of K246 residue).
DOI: https://doi.org/10.7554/eLife.32952.013

---

of GSK3α. Further, our MS/MS analysis indicate that K99 and K246 are the acetylation sites in GSK3α (*Figure 4D*). The protein sequence alignment of GSK3β and GSK3α showed that K246 of GSK3α is analogous to residue K183 of GSK3β (*Figure 4E*). To further strengthen our assumption, we generated the homology model of GSK3α using the online Swiss-Modeler tool (*Biasini et al., 2014*). The modeled GSK3α further confirmed that the residue K246 would occupy the same position proximal to the adenine nucleotide as K183 of GSK3β (*Figure 4F*). We therefore assume that acetylation of residue K246 would play a similar role in de-stabilizing adenine nucleotide in the pocket of GSK3α.

## GSK3 is required for the anti-hypertrophic role of SIRT2 deacetylase

Activation of GSK3β antagonizes the development of cardiac hypertrophy (*Antos et al., 2002*; *Kerkela et al., 2008*; *Sugden et al., 2008*). Cardiac myocyte-specific deletion of GSK3α and GSK3β, together, results in severe dilated cardiomyopathy (*Zhou et al., 2016*). Recently, SIRT2 was demonstrated to be an anti-hypertrophic molecule (*Tang et al., 2017*). In our recent work, we also found that SIRT2 deficiency induces spontaneous cardiac hypertrophy in mice by hyperactivation of NFAT transcription factors (*Sarikhani et al., 2018a*). To understand the contribution of GSK3 as a key molecule in the anti-hypertrophic signaling regulated by SIRT2, we treated SIRT2 overexpressing cardiomyocytes with GSK3 inhibitors, LiCl (*Bertsch et al., 2011*) and GSK3 inhibitor X (*Li et al., 2014*; *Meijer et al., 2003*) and studied the ISO-induced cardiac hypertrophy. Treatment with GSK3 inhibitors markedly reduced the activity of GSK3 as measured by phosphorylation of GS (*Figure 5A*). In cardiomyocytes, SIRT2 overexpression reduces ISO-induced protein synthesis, cardiac myocyte size and the expression of ANP, a fetal gene that is considered as a marker of cardiac hypertrophy. However, treatment of GSK3 inhibitors abrogated the anti-hypertrophic role of SIRT2 in cardiomyocytes treated with ISO (*Figure 5B–D*, *Figure 5—figure supplement 1*), suggesting that both GSK3 isoforms are required for the anti-hypertrophic function of SIRT2 deacetylase.

The acetyl transferase p300 enhances the acetylation of GSK3 and thus inhibits its catalytic activity. Therefore, we tested whether depletion of p300 could reduce the acetylation of GSK3 and thus rescue the effect of SIRT2 deficiency. Our *in vitro* global protein synthesis experiment, which measures the incorporation of puromycin (*Schmidt et al., 2009*), suggest that depletion of p300 reduces the abnormal protein synthesis in SIRT2-depleted cardiomyocytes (*Figure 5E and F*). To validate our findings *in vivo*, we performed rescue experiments with anacardic acid, a p300 inhibitor. Western blotting analysis suggested that treatment with anacardic acid markedly reduces GSK3β acetylation in SIRT2-deficient mice (*Figure 5G and H*). Furthermore, we found that treatment of anacardic acid partially restores the activity of GSK3 in SIRT2-deficient hearts, as measured by phosphorylation of GS (*Figure 5G*), indicating that p300 inhibition could rescue the impaired activity of GSK3β caused by SIRT2 deficiency. However, the change in acetylation status of GSK3β after p300 inhibitor treatment is not correlated well with the GSK3β activity on glycogen synthase. We believe that glycogen synthase might have been post-translationally modified in SIRT2-KO hearts. It is possible that SIRT2-KO hearts might have increased expression of phosphatases that could regulate glycogen synthase phosphorylation. In our recent work, we demonstrated the development of spontaneous cardiac hypertrophy in SIRT2-KO mice (*Sarikhani et al., 2018a*). In the present study, we found similar

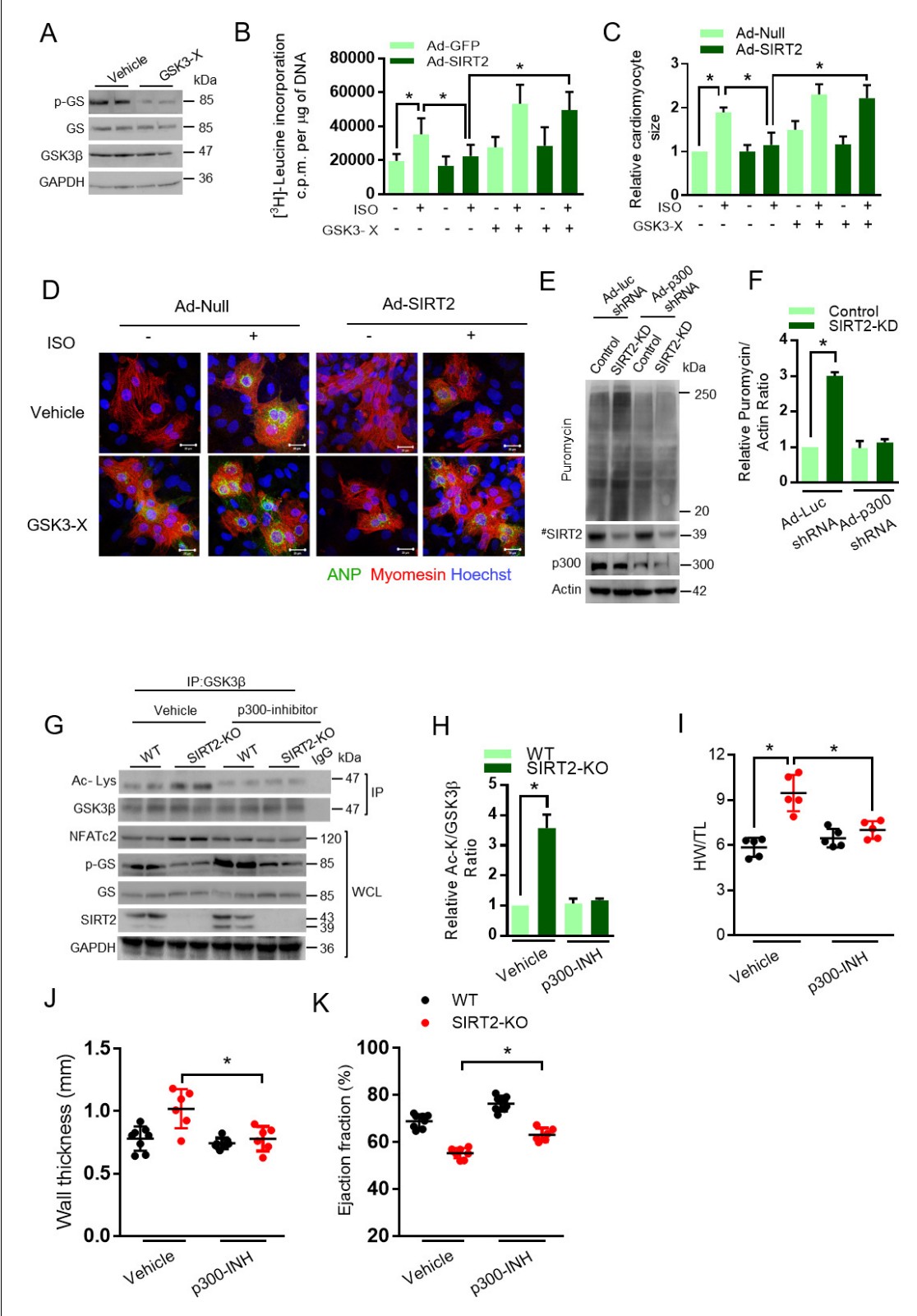

**Figure 5.** GSK3β is required for the anti-hypertrophic role of SIRT2 deacetylase. (**A**) Western blotting analysis depicting the activity of GSK3 inhibitor X (GSK3-X). Neonatal rat cardiomyocytes were treated with vehicle or 500 nM GSK3-X for 48 hr and the activity of GSK3 was assessed by monitoring the phosphorylation of GS by specific antibody. (**B**) [³H]-leucine incorporation into total cellular protein of control (Ad-GFP) or SIRT2-overexpressing (Ad-SIRT2) rat neonatal cardiomyocytes treated with either vehicle or 500 nM GSK3 inhibitor X (GSK3-X) for 48 hr. Cardiomyocytes were infected with

*Figure 5 continued on next page*

*Figure 5 continued*

adenoviral vectors encoding either GFP or SIRT2 for 24 hr prior to GSK3-X treatment. After the GSK3-X treatment, cardiomyocytes were stimulated with either vehicle or 20 µM ISO for 24 hr and the [$^3$H]-leucine incorporation was monitored. c.p.m. counts per minute. $n$ = 10. Data is presented as mean ± s.d. *p<0.05. Two-way ANOVA was used to calculate the p values. (C) Histogram showing quantification of relative cardiomyocyte area in control (Ad-Null) and SIRT2-overexpressing (Ad-SIRT2) rat neonatal cardiomyocytes treated with either vehicle or 500 nM GSK3 inhibitor X (GSK3-X) for 48 hr. Cardiomyocytes were infected with adenoviral vectors encoding either control or SIRT2 for 24 hr prior GSK3-X treatment. After the GSK3-X treatment, cardiomyocytes were stimulated with either vehicle or 20 µM ISO for 24 hr and the relative cardiomyocyte area is quantified as described in Materials and methods section. Data is presented as mean ± s.d. *p<0.05. Two-way ANOVA was used to calculate the p values. (D) Representative confocal images depicting perinuclear expression of ANP in control (Ad-Null) or SIRT2-overexpressing (Ad-SIRT2) cardiomyocytes treated with either vehicle or ISO (20 µM, 24 hr), with or without GSK3 inhibitor X (GSK3-X, 500 nM, 48 hr). Scale bar = 20 µm. ANP (Green), Myomesin (Red), Hoechst (Blue). (E) Western blotting analysis for puromycin incorporation in control or SIRT2-depleted (SIRT2-KD) neonatal rat cardiomyocytes infected with adenovirus expressing either control (Ad-luc shRNA) or p300 shRNA (Ad-p300-shRNA) 48 hr. p300 depletion was confirmed by western blotting. Pulse of puromycin was given 30 min prior to harvesting of cardiomyocytes and puromycin incorporation into nascent proteins was tested using anti-puromycin antibody. # marked Western images denotes SIRT2 antibody used in this assay detects single band. (F) Histogram showing relative puromycin levels in control or SIRT2-depleted (SIRT2-KD) cardiomyocytes infected with adenovirus expressing either control (Ad-luc shRNA) or p300 shRNA (Ad-p300-shRNA). The data is generated from *Figure 5E*. Signal intensities of puromycin and actin were measured by densitometry analysis using ImageJ software. $n$ = 3 independent experiments. Data is presented as mean ± s.d. *p<0.05. Two-way ANOVA was used to calculate the p values. (G) Western blotting analysis of GSK3β acetylation and activity in heart lysates of vehicle or anacardic acid (p300 inhibitor) treated 9 months old WT and SIRT2-KO mice littermates. Anacardic acid was injected intraperitoneal at the dose of 5 mg/kg/day for 10 days in mice. Peanut oil was used as vehicle. GSK3β was immunoprecipitated from heart lysates of WT and SIRT2-KO mice using anti-GSK3β antibody (sc-9166, Santa Cruz Biotechnology), and the affinity resin with protein A/G immobilized. Western blotting was performed to detect GSK3β acetylation by anti-Ac-Lysine antibody. GSK3β activity was measured by detecting the phosphorylation of GS. SIRT2 depletion was confirmed by western blotting. Whole cell lysates (WCL) was probed for indicated proteins by western blotting. (H) Histogram showing relative GSK3β acetylation in heart lysates of vehicle or anacardic acid (5 mg/kg/day for 10 days) treated 9 months old WT and SIRT2-KO mice from *Figure 5G*. $n$ = 3. Signal intensities of GSK3β and acetylated-GSK3β was measured by densitometry analysis using ImageJ software. Data is presented as mean ± s.d. *p<0.05. Two-way ANOVA was used to calculate the p values. (I) Scatter plot depicting HW/TL ratio of 9 months old WT and SIRT2-KO mice treated with either vehicle or anacardic acid, (p300-INH), at the dose of 5 mg/kg/day for 10 days. $n$ = 5 mice per group. Data is presented as mean ± s.d. *p<0.05. Two-way ANOVA was used to calculate the p values. (J) Scatter plot showing left ventricular posterior wall thickness of 9 months old WT and SIRT2-KO mice treated with either vehicle or anacardic acid (p300-INH), at the dose of 5 mg/kg/day for 10 days. $n$ = 6–8 mice per group. Data is presented as mean ± s.d. *p<0.05. Two-way ANOVA was used to calculate the p values. (K) Scatter plot depicting cardiac contractile functions, as measured by ejection fraction of 9 months old WT and SIRT2-KO mice treated with either vehicle or anacardic acid (p300-INH), at the dose of 5 mg/kg/day for 10 days. $n$ = 6–8 mice per group. Data is presented as mean ± s.d. *p<0.05. Two-way ANOVA was used to calculate the p values.

DOI: https://doi.org/10.7554/eLife.32952.014

The following figure supplement is available for figure 5:

**Figure supplement 1.** GSK3 inhibition abrogates anti-hypertrophic role of SIRT2 deacetylase.

DOI: https://doi.org/10.7554/eLife.32952.015

results as SIRT2 deficiency induces spontaneous cardiac hypertrophy in mice (*Figure 5I–J*). Interestingly, inhibition of p300 significantly reduced HW/TL ratio, and wall thickness, while increasing ejection fraction in SIRT2-deficient mice (*Figure 5I–K*), indicating that p300 inhibition rescues cardiac hypertrophy and contractile dysfunction resulting from SIRT2 deficiency, which could be due to the restoration of GSK3 activity.

## Discussion

We identified acetylation of GSK3 isoforms as novel post-translational modification, which hinders the ATP binding, and thus inhibits its kinase activity independent of its inhibitory phosphorylation during heart failure. We found that SIRT2 binds to and deacetylates GSK3β at Lys183, which enhances ATP binding and its kinase activity.

The highly conserved lysine residues in the ATP-binding pocket that is Lys183 in GSK3β and Lys246 in GSK3α, are acetylated. The acetylated lysine residues interfere with the binding of ATP to GSK3 due to charge repulsion and thus reduce the catalytic activity. We believe that the mechanism of inhibition of GSK3 isoforms by acetylation is unique when compared to inhibitory phosphorylation, where the phosphorylation of GSK3α at Ser21 or GSK3β at Ser9 causes the N-terminal tail of GSK3 to act as a pseudo substrate (*Dajani et al., 2001*; *Stamos et al., 2014*). The phosphorylated serine tail hinders the binding of primed substrates by self-associating with the primed-substrate binding pocket of GSK3, and thus reduce the kinase activity of GSK3 (*Dajani et al., 2001*;

*Stamos et al., 2014*). It is worth mentioning that acetylation might even inhibit the activity of non-phosphorylated GSK3, which is capable of binding to the primed-substrates. On the other hand, deacetylated GSK3 can also be inhibited by Ser9 phosphorylation, since both acetylation and phosphorylation are two independent events regulating the activity of GSK3 isoforms. In contrast to our findings, it has previously been demonstrated that treatment of AGK2 in platelets decreases Ser9 phosphorylation of GSK3β, due to enhanced acetylation and reduced activity of the Akt, an upstream kinase capable of phosphorylating GSK3β at Ser9 residue (*Moscardó et al., 2015*), suggesting that the relationship between SIRT2 and Ser9 phosphorylation of GSK3β is context-dependent. In our previous work, we found that deacetylation of Akt by SIRT1 enhances its kinase activity and thus SIRT1 indirectly inhibits GSK3 activity by increasing Ser9 phosphorylation (*Sundaresan et al., 2011*). Our current work demonstrates that deacetylation of GSK3 by SIRT2 promotes its activity. Consequently, acetylation is not only involved with direct regulation of GSK3, but also by an indirect mechanism via PDK1/AKT signaling.

In our previous work, we identified acetylation of GSK3β at Lys15, which influences its mitochondrial localization (*Sundaresan et al., 2016*). Since Lys15 is away from the ATP-binding pocket of GSK3β, we assume that Lys15 will not regulate the ATP binding. Similarly, it has previously been shown that K189 in the Arabidopsis GSK3 orthologue BIN2 (equivalent to K205 in human GSK3β) is targeted by the deacetylase HDAC6 and the deacetylation of K189 in BIN2 was shown to decrease its kinase activity (*Hao et al., 2016*). Our study suggests that the acetylation of K183 decreases GSK3β kinase activity by directly interfering with its adenine nucleotide binding. Since, K189 in the Arabidopsis GSK3 orthologue BIN2 in activation loop located away from the nucleotide-binding site (data not shown), it would certainly follow a distinct mechanism in influencing the kinase activity of BIN2. Therefore, we believe that distinct acetylation events within the kinase domain of GSK3 might have opposing effects on its kinase activity.

Previous studies indicate that Tyr216 phosphorylation of GSK3β is dynamic and mirrors its catalytic activity. Tyr216 phosphorylation of GSK3β has been shown to decrease immediately after reoxygenation in cultured cortical neurons (*Chen et al., 2016*). In contrast, the phosphorylation of GSK3β at Tyr216 residue increases during ischemic injury of heart (*Abdillahi et al., 2012*) and after IFN-γ treatment in RAW264.7 cells (*Tsai et al., 2009*). Interestingly, phosphorylation of GSK3β at Tyr216 was frequently increased and was predictive of better prognosis in early-stage gastric carcinoma (*Cho et al., 2010*). Studies indicate that Tyr216 phosphorylation of GSK3β is required for its maximal activity and is considered as an autophosphorylation event that occurs during protein translation (*Dajani et al., 2001*; *Hughes et al., 1993*). Moreover, the phosphorylation of the endogenous GSK3β at Tyr216 residue was suppressed by small molecule inhibitors of GSK3, which are known to block autophosphorylation at Tyr216 residue and catalytic activity of GSK3β (*Cole et al., 2004*). Our work demonstrates that the phosphorylation of GSK3β at Tyr216 residue was reduced in SIRT2-deficient mice. Since acetylation hinders the binding of ATP to GSK3β, it is possible that autophosphorylation of GSK3β might also be reduced in SIRT2-deficient conditions, when GSK3β is mostly acetylated. Conventionally, GSK3β is considered to be a cytoplasmic protein, although it is found inside nucleus and mitochondria (*Sundaresan et al., 2016*). However, the p300 is mostly nuclear protein and SIRT2 is majorly a cytoplasmic protein. Since, GSK3 shuttles between cytoplasm and nucleus, it is possible that p300 and SIRT2 may function as acetyl transferase and deacetylase respectively, although they are localized in different subcellular compartments. During cellular stress, GSK3β localizes to mitochondria and this localization is required for cellular apoptosis (*Maurer et al., 2006*). In our previous work, we identified SIRT3-deacetylase-mediated regulation of Lys15 to be critical for its mitochondrial localization (*Sundaresan et al., 2016*). However, the current work suggests that SIRT2-dependent regulation of GSK3β Lys183 acetylation do not influence its cytoplasmic localization. Since Sirtuins are localized to sub-cellular compartments, SIRT2 might regulate the GSK3β activity in the cytoplasm, while SIRT3 is controlling the GSK3β activity inside mitochondria.

GSK3 plays key physiological roles in cellular protein synthesis, cell survival, cell proliferation, cell differentiation, cellular metabolism, microtubule dynamics and cell motility (*Beurel et al., 2015*; *Cohen and Frame, 2001*; *Doble and Woodgett, 2003*; *Rayasam et al., 2009*). Impaired GSK3 activity is involved in the development of diverse pathological conditions like cardiovascular disease, obesity, inflammation, diabetes, Alzheimer disease and cancer (*Cohen and Frame, 2001*). GSK3 is one of the few signaling mediators that organize diverse signaling pathways, including those

activated by Wnts, growth factors, cytokines, G protein-coupled, and α and β adrenoceptor ligands (*Sugden et al., 2008*). GSK3 isoforms play a key role in antagonizing the development of cardiac hypertrophy. GSK3 also blocks cellular protein synthesis necessary for the development of cardiac hypertrophy (*Sugden et al., 2008*). In particular, active GSK3 prevents cardiac hypertrophic growth and fibrosis through inhibition of transcription factors such as NFAT, GATA4, β-Catenin, myocardin, c-Myc, and c-Jun, and the translational regulator, eIF2Bε (*Sugden et al., 2008*). There are reports suggesting that endothelin-1, GPCR, and β-adrenoceptor ligands induce N-terminal serine phosphorylation of GSK3 in cardiomyocytes, that might be one of the possible mechanism of GSK3 inhibition in the heart (*Matsuda et al., 2008*; *Sugden et al., 2008*). On the other hand, studies conducted on mice models suggest that N-terminal Ser9 phosphorylation of GSK-3β was neither changed, nor is associated with the reduced activity in the heart after pressure overload or myocardial infarction (*Zhai et al., 2007*). On similar lines, GSK3 S21A/S9A knock-in mice, which is resistant to inhibitory N-terminal phosphorylation is not protected from cardiac hypertrophy, suggesting that inhibitory N-terminal phosphorylation is not the only mechanism that regulates the activity of GSK3 (*Matsuda et al., 2008*).

Our work suggests that GSK3 activity is critically regulated by phosphorylation-independent mechanisms such as acetylation during the heart failure. Interestingly, inhibition of the acetyl transferase p300 reduced the acetylation and partially restored the activity of GSK3β in SIRT2-deficient cells. Furthermore, treatment of GSK3 inhibitors attenuate the beneficial effects of SIRT2 overexpression, indicating the importance of GSK3 in SIRT2-mediated regulation of cardiac hypertrophy. A recent work suggests that SIRT2 deacetylates LKB1 to promote AMPK activity and regulates the development of cardiac hypertrophy (*Tang et al., 2017*). In our recent work, we also found that SIRT2 deficiency induces spontaneous cardiac hypertrophy in mice through hyperactivation of NFAT transcription factors (*Sarikhani et al., 2018a*). In the current work, we found that GSK3β activity is required for the protective functions of SIRT2 in cardiomyocytes. Previous studies indicate that GSK3β is a master regulator of divergent signaling pathways. Notably, GSK3β phosphorylates and represses NFAT transcription factors (*Neal and Clipstone, 2001*). Similarly, GSK3β and AMPK are known to regulate each other (*Horike et al., 2008*; *Suzuki et al., 2013*). It is possible that SIRT2 might have multiple and redundant targets to regulate cardiac hypertrophy, as observed in members of sirtuin family (*Martínez-Redondo and Vaquero, 2013*).

Overall, our work identified GSK3 isoforms as a novel target of SIRT2 deacetylase. SIRT2-dependent regulation critically modulates the activity of GSK3β independent of Ser9 inhibitory phosphorylation.

# Materials and methods

## Key resources table

| Reagent type (species) or resources | Designation | Source or reference | Identifiers | Additional information |
|---|---|---|---|---|
| Strain, A2:A128 strain background (*Mus musculus*, C57BL/6J) | Sirt2 knockout mice, JAX Stock #012772 - B6.129-Sirt2 < tm1.1Fwa>/J | Jackson Laboratories, USA | | |
| Strain, strain background (*Mus musculus*, 129/SvJ) | WT, JAX stock # 000691 | Jackson Laboratories, USA | | |
| Cell line (human) | HeLa | ATCC | | |
| Cell line (human) | HEK 293 | ATCC | | |
| Cell line (mouse) | GSK3β-KO fibroblasts | James Woodgett, Mount Sinai Hospital, Toronto, Canada | | |
| Strain, (Wistar rats) | WT | Central Animal Facility, Indian Institute of Science, India | | P1-P2 pups used for primary cardiomyocytes culture |

*Continued on next page*

*Continued*

| Reagent type (species) or resources | Designation | Source or reference | Identifiers | Additional information |
|---|---|---|---|---|
| Antibody | anti-GSK3β | Cell Signaling Technology | 9315 | 1:1000 diluted in 5% BSA |
| Antibody | anti-GSK3β | Santa Cruz Biotechnology | sc-9166 | 1:1000 diluted in 5% milk for Western blotting 1:200 diluted in 1% BSA for immuno-fluorescence |
| Antibody | anti-Acetylated-Lysine | Cell Signaling Technology | 9681 | 1:1000 diluted in 5% BSA |
| Antibody | anti-Acetylated-Lysine | Cell Signaling Technology | 9441 | 1:1000 diluted in 5% BSA |
| Antibody | anti-GSK3β Ser-9 | Cell Signaling Technology | 9336 | 1:1000 diluted in 5% BSA |
| Antibody | anti-GSK3β Tyr 279/216 | Merck Millipore | 05–413 | 1:250 diluted in 5% BSA |
| Antibody | anti-Phospho-β-Catenin | Cell Signaling Technology | 9561 | 1:2000 diluted in 5% BSA |
| Antibody | anti-β-Catenin | Cell Signaling Technology | 8480 | 1:1000 diluted in 5% BSA |
| Antibody | anti-SIRT2 | Sigma-Aldrich | S8447 | 1:2000 diluted in 5% BSA |
| Antibody | anti-SIRT2 | Merck Millipore | 09–843 | 1:1000 diluted in 5% BSA |
| Antibody | anti-SIRT2 | Cell Signaling Technology | 12650 | 1:1000 diluted in 5% BSA |
| Antibody | anti-ANP | Abcam | 14348 | 1:250 diluted in 5% BSA |
| Antibody | anti-ANP | Cloud-Clone Corporation | PAA225Ra03 | 1:200 diluted in 1% BSA |
| Antibody | anti-GAPDH | Santa Cruz Biotechnology | sc-25778 | 1:1000 diluted in 5% milk |
| Antibody | anti-p300 | Merck Millipore | 05–257 | 1:1000 diluted in 5% BSA for Western blotting, 1:200 diluted in 1% BSA for immuno-fluorescence |
| Antibody | anti-phospho-Glycogen Synthase | Merck Millipore | 07–817 | 1:1000 diluted in 5% BSA |
| Antibody | anti-Glycogen Synthase | Cell Signaling Technology | 3893 | 1:1000 diluted in 5% BSA |
| Antibody | anti-β -actin (HRP-conjugate) | Cell Signaling Technology | 12262 | 1:3000 diluted in 5% BSA |
| Antibody | anti-β -actin (HRP-conjugate) | Sigma-Aldrich | A3854 | 1:3000 diluted in 5% BSA |
| Antibody | anti-GSK3 α/β | Merck Millipore | 04–903 | 1:1000 diluted in 5% BSA |
| Antibody | anti-puromycin | Developmental Studies Hybridoma Bank | PMY-2A4 | 1:500 diluted in 5% BSA |
| Antibody | anti-NFATc2 | Thermo Fisher Scientific | MA1-025 | 1:100 diluted in 5% BSA |
| Antibody | anti-Flag | Sigma-Aldrich | F2555 | 1:2000 diluted in 5% BSA |
| Antibody | anti-α-Tubulin | Cell Signaling Technology | 2144 | 1:2000 diluted in 5% BSA |
| Antibody | anti-Acetyl-α-Tubulin (Lys40) | Cell Signaling Technology | 5335 | 1:1000 diluted in 5% BSA |
| Antibody | anti-SIRT1 | Santa Cruz Biotechnology | sc-15404 | 1:1000 diluted in 5% milk |
| Antibody | anti-SIRT3 | Cell Signaling Technology | 5490 | 1:1000 diluted in 5% BSA |
| Antibody | anti-SIRT4 | Cloud-Clone Corporation | PAE914Hu01 | 1:500 diluted in 5% BSA |
| Antibody | anti-SIRT5 | Cloud-Clone Corporation | PAE915Mu01 | 1:500 diluted in 5% BSA |
| Antibody | anti-SIRT6 | Cell Signaling Technology | 12486 | 1:1000 diluted in 5% BSA |
| Antibody | anti-SIRT7 | Cloud-Clone Corporation | PAE917Hu01 | 1:500 diluted in 5% BSA |
| Antibody | anti-HA | Sigma-Aldrich | H9658 | 1:2000 diluted in 5% BSA |
| Antibody | anti-HA | Santa Cruz Biotechnology | sc-805 | 1:100 diluted in 1% BSA for immuno-fluorescence |
| Antibody | anti-p300 | Merck Millipore | 05–257 | 1:1000 diluted in 5% BSA for Western, 1:100 diluted in 1% BSA for immuno-fluorescence |
| Antibody | anti-SOD2 | Santa Cruz Biotechnology | sc-515068 | 1:200 diluted in 5% milk |
| Antibody | anti-α -Actinin | Sigma-Aldrich | A5044 | 1:200 diluted in 5% BSA |
| Antibody | Clean-Blot IP Detection Reagent | Thermo Fisher Scientific | 21230 | 1:2000–5000 diluted in 5% milk |

*Continued on next page*

*Continued*

| Reagent type (species) or resources | Designation | Source or reference | Identifiers | Additional information |
|---|---|---|---|---|
| Antibody | anti-rabbit HRP | Santa Cruz Biotechnology | sc-2004 | 1:5000 diluted in 1% milk |
| Antibody | anti-mouse HRP | Santa Cruz Biotechnology | sc-2005 | 1:5000 diluted in 1% milk |
| Antibody | anti-mouse HRP | Thermo Fisher Scientific | 31430 | 1:5000 diluted in 1% milk |
| Antibody | anti-rabbit HRP | Thermo Fisher Scientific | 31460 | 1:5000 diluted in 1% milk |
| Antibody | anti-rabbit IgG light chain HRP | Abcam | ab99697 | 1:5000 diluted in 1% milk |
| Antibody | Donkey anti-mouse, Alexa Fluor 488 | Thermo Fisher Scientific | A-21202 | 1:200 diluted in 5% BSA |
| Antibody | Goat anti-rabbit, Alexa Fluor 546 | Thermo Fisher Scientific | A-11035 | 1:200 diluted in 5% BSA |
| Antibody | Ni-NTA Agarose | Qiagen | 30230 | |
| Antibody | ANTI-FLAG M2 Affinity Agarose Gel | Sigma-Aldrich | A2220 | |
| Antibody | Glutathione Sepharose 4B | GE healthcare | 17-0756-01 | |
| Antibody | Monoclonal Anti-HA−Agarose antibody produced in mouse | Sigma-Aldrich | A2095 | |
| Antibody | Protein A/G Agarose | Santa Cruz Biotechnology | sc-2003 | |
| Transfected construct | pcDNA3 Flag HA | Addgene | Plasmid 10792 | 1436 pcDNA3 Flag HA plasmid DNA was a gift from William Sellers |
| Transfected construct (human) | HA GSK3 beta wt pcDNA3 | Addgene | Plasmid 14753 | PMID: 7715701 |
| Transfected construct (human) | HA GSK3 alpha wt | Modified Addgene, plasmid 15896 | This paper | |
| Transfected construct (human) | HA GSK3 beta S9A pcDNA3 | Addgene | Plasmid 14754 | PMID: 7980435 |
| Transfected construct (human) | HA GSK3 beta K85A pcDNA3 | Addgene | Plasmid 14755 | HA GSK3 beta K85A pcDNA3 was a gift from Jim Woodgett |
| Transfected construct (human) | HA GSK3 beta K150Q pcDNA3 | Modified Addgene, plasmid 14753 | This paper | For, caccggcagggtctgctgcgcgcg gctataatg; Rev, cattatagccgcgc gcagcagaccctgccggtg; |
| Transfected construct (human) | HA GSK3 beta K150R pcDNA3 | Modified Addgene, plasmid 14754 | This paper | For, attatagccgcgcgagacagac cctgccg; Rev, cggcagggtctgtc tcgcgcggctataat; |
| Transfected construct (human) | HA GSK3 beta K183Q pcDNA3 | Modified Addgene, plasmid 14755 | This paper | For, gcaggttctgcggctgaatatcgcg atggcaaatgccaaag; Rev, ctttggcatttgcc atcgcgatattcagccgcagaacctgc; |
| Transfected construct (human) | HA GSK3 beta K183R pcDNA3 | Modified Addgene, plasmid 14756 | This paper | For, aggttctgcggtctaatatcgc gatggcaaatgcca; Rev, tggcatttgccat cgcgatattagaccgcagaacct. |
| Transfected construct (human) | GST-GSK3β | | This paper | |
| Transfected construct (human) | HIS- GSK3β | | This paper | |
| Transfected construct (human) | HIS- GSK3β-K183R | | This paper | |
| Transfected construct (human) | HIS- GSK3β-K183Q | | This paper | |
| Transfected construct (human) | SIRT1 Flag | Addgene | Plasmid 13812 | PMID: 12620231 |
| Transfected construct (human) | SIRT2 Flag | Addgene | Plasmid 13813 | PMID: 12620231 |

*Continued on next page*

*Continued*

| Reagent type (species) or resources | Designation | Source or reference | Identifiers | Additional information |
|---|---|---|---|---|
| Transfected construct (human) | SIRT3 Flag | Addgene | Plasmid 13814 | PMID: 12620231 |
| Transfected construct (human) | SIRT4 Flag | Addgene | Plasmid 13815 | PMID: 12620231 |
| Transfected construct (human) | SIRT5 Flag | Addgene | Plasmid 13816 | PMID: 12620231 |
| Transfected construct (human) | SIRT6 Flag | Addgene | Plasmid 13817 | PMID: 12620231 |
| Transfected construct (human) | SIRT7 Flag | Addgene | Plasmid 13818 | PMID: 12620231 |
| Transfected construct (human) | SIRT2-H187Y Flag | Modified from Addgene, plasmid 13818 | | For 5'atgtgtagaaggtgccat acgcctccaccaagtcc3'- Rev 5' ggacttggtggaggcgtatgg caccttctacacat3'. |
| Infected construct (human) | Ad-Null | Vector Biolabs | Adenovirus 1300 | |
| Infected construct (human) | Ad-GFP | Vector Biolabs | Adenovirus 1060 | |
| Infected construct (human) | Ad-SIRT2 | Vector Biolabs | Adenovirus 1519 | |
| Infected construct (human) | Ad-h-EP300 | Vector Biolabs | Adenovirus ADV-207954 | |
| Infected construct (human) | Ad-luc-shRNA | B. Thimmapaya, Northwestern University, Chicago, IL, USA | | PMID: 26667039 |
| Infected construct (human) | Ad-h-EP300-shRNA | B. Thimmapaya, Northwestern University, Chicago, IL, USA | | PMID: 26667039 |
| Recombinant protein | p300 | Merck Millipore | 14–418 | http://dx.doi.org/10.1038/s41418-018-0069-8 |
| Sequence-based reagent | SMART pool: siGENOME Non-Targeting siRNA 1 | Dharmacon | D-001206-13-50 | 100 nM siRNA transfected by Lipofectamine RNAiMAX Transfection Reagent |
| Sequence-based reagent | SMART pool: siGENOME Rat Sirt2 siRNA | Dharmacon | M-082072-01-0010 | 100 nM SMARTpool siRNA transfected by Lipofectamine RNAiMAX Transfection Reagent |
| Commercial assay or kit | GSK-3 Activity Assay Kit | Sigma-Aldrich | CS0990 | PMID: 26667039 |
| Commercial assay or kit | QuikChange Site-Directed Mutagenesis Kit | Agilent Technologies | 200518 | PMID: 26667039 Sequences verified by sequencing, SciGenom Labs |
| Commercial assay or kit | GenElute HP Plasmid Midiprep Kit | Sigma-Aldrich | NA0200 | |
| Commercial assay or kit | GEnElute HP Plasmid Miniprep Kit | Sigma-Aldrich | PLN70 | |
| Commercial assay or kit | Qubit dsDNA HS assay kit | Thermo Fisher Scientific | Q32851 | PMID: 25871545 |
| Chemical compound, drug | Lipofectamine 2000 Transfection Reagent | Thermo Fisher Scientific | 11668019 | |
| Chemical compound, drug | Lipofectamine RNAiMAX Transfection Reagent | Thermo Fisher Scientific | 13778150 | |
| Chemical compound, drug | Horse serum, heat inactivated | Thermo Fisher Scientific | 26050088 | |
| Chemical compound, drug | Fetal Bovine Serum | Thermo Fisher Scientific | 10500064 | |

*Continued on next page*

*Continued*

| Reagent type (species) or resources | Designation | Source or reference | Identifiers | Additional information |
|---|---|---|---|---|
| Chemical compound, drug | Penicillin-Streptomycin | Thermo Fisher Scientific | 15070063 | |
| Chemical compound, drug | Gelatin, Type B | Sigma-Aldrich | G9382 | 0.2% w/v |
| Chemical compound, drug | D-glucose | Sigma-Aldrich | G8270 | 0.01 M prepared in PBS |
| Chemical compound, drug | Collagenase, Type II | Thermo Fisher Scientific | 17101015 | 0.4 mg/ml prepared in Trypsin-PBS-Glucose |
| Chemical compound, drug | Trypsin | Thermo Fisher Scientific | 15050057 | 0.2% prepared in PBS-Glucose |
| Chemical compound, drug | Trypsin-EDTA | Thermo Fisher Scientific | 25200056 | 0.1% prepared in PBS |
| Chemical compound, drug | Isoproterenol | Sigma-Aldrich | I6504 | https://doi.org/10.1172/JCI39162 |
| Chemical compound, drug | AGK2 | Cayman Chemical | 13145 | http://dx.doi.org/10.1038/s41418-018-0069-8 |
| Chemical compound, drug | Lithium chloride | Sigma-Aldrich | 203637 | PMID: 20926980 |
| Chemical compound, drug | GSK-3 Inhibitor X | Calbiochem | CAS 740841-15-0 - | PMID: 16984885 |
| Chemical compound, drug | Anacardic Acid | Cayman Chemical | CAS16611840 | PMID: 28513807 |
| Chemical compound, drug | Puromycin | VWR | J593 | |
| Chemical compound, drug | Fluoromount-G | Southern Biotech | 0100–01 | |
| Chemical compound, drug | Hoechst 33342 | Thermo Fisher Scientific | H3570 | |
| Chemical compound, drug | Dulbecco's Modified Eagle's Medium- High glucose | Sigma-Aldrich | D5648 | |
| Chemical compound, drug | Isoflurane | Sosrane Neon Laboratories Ltd | | |
| Chemical compound, drug | Nicotinamide | Sigma-Aldrich | N3376 | |
| Chemical compound, drug | Trichostatin A | Sigma-Aldrich | T8552 | |
| Chemical compound, drug | ProLong Gold Antifade Mounting medium with DAPI | Thermo Fisher Scientific | P36931 | |
| Chemical compound, drug | Acrylamide | Sigma-Aldrich | A9099 | |
| Chemical compound, drug | Tris | Sigma-Aldrich | T6066 | |
| Chemical compound, drug | Hydrochloric acid | Fischer Scientific | 29505 | |
| Chemical compound, drug | Sodium-dodecyl sulphate | VWR | 0227 | |
| Chemical compound, drug | Ammonium persulphate | Sigma-Aldrich | A3678 | |
| Chemical compound, drug | TEMED | Sigma-Aldrich | T7024 | |
| Chemical compound, drug | Sodium chloride | Merck Millipore | 106404 | |

*Continued on next page*

Continued

| Reagent type (species) or resources | Designation | Source or reference | Identifiers | Additional information |
|---|---|---|---|---|
| Chemical compound, drug | Triton X-100 | Sigma-Aldrich | T8787 | |
| Chemical compound, drug | EDTA | Sigma-Aldrich | E5134 | |
| Chemical compound, drug | EGTA | Sigma-Aldrich | 324626 | |
| Chemical compound, drug | sodium pyrophosphate | Sigma-Aldrich | 221368 | |
| Chemical compound, drug | sodium orthovanadate | Sigma-Aldrich | 450243 | |
| Chemical compound, drug | Tween-20 | Sigma-Aldrich | P9416 | |
| Chemical compound, drug | cOmplete, Mini Protease Inhibitor Cocktail | Sigma-Aldrich | 11836153001 ROCHE | |
| Chemical compound, drug | PMSF | Sigma-Aldrich | P7626 | |
| Chemical compound, drug | 2X Laemmli Sample Buffer | Bio-Rad | 161–0737 | |
| Chemical compound, drug | β-mercaptoethanol | VWR | 0482 | |
| Chemical compound, drug | DMSO | Sigma-Aldrich | D8418 | |
| Chemical compound, drug | Clarity ECL Western Blotting Substrate | BioRad | 5060 | |
| Chemical compound, drug | SuperSignal West Pico chemiluminescent Substrate | Thermo Fisher Scientific | 34080 | |
| Chemical compound, drug | IPTG | Sigma-Aldrich | I6758 | |
| Chemical compound, drug | Glycerol | PUREGENE | PG-4580 | |
| Chemical compound, drug | Sodium hydroxide | Sigma-Aldrich | 221465 | |
| Chemical compound, drug | Calcium chloride | Sisco Research Laboratories | 70650 | |
| Chemical compound, drug | formaldehyde solution | Sigma-Aldrich | F1635 | |
| Chemical compound, drug | Bovine serum albumin | HIMEDIA | MB083 | |
| Chemical compound, drug | Glycine | Fischer Scientific | 12835 | |
| Chemical compound, drug | Non-fat-milk | HIMEDIA | GRM1254 | |
| Chemical compound, drug | Methanol | Honeywell | 230–4 | |
| Chemical compound, drug | Bis-acrylamide | Sigma-Aldrich | M7279 | |
| Chemical compound, drug | Sodium deoxycholate | Sigma-Aldrich | D6750 | |
| Chemical compound, drug | Sodium bicarbonate | Sigma-Aldrich | S6014 | |
| Chemical compound, drug | Bio-Rad Protein Assay Dye Reagent Concentrate | Bio-Rad | 5000006 | |

Continued

| Reagent type (species) or resources | Designation | Source or reference | Identifiers | Additional information |
|---|---|---|---|---|
| Chemical compound, drug | Ampicillin | VWR | 0339 | |
| Chemical compound, drug | Leucine-free minimal essential medium | Thermo fisher Scientific | 30030 | |
| Chemical compound, drug | DTT | Sigma-Aldrich | DTT-RO | |
| Chemical compound, drug | Sodium butyrate | Sigma-Aldrich | 567430 | |
| Chemical compound, drug | Magnesium chloride | Sigma-Aldrich | M8266 | |
| Chemical compound, drug | Sodium fluoride | Sigma-Aldrich | 450022 | |
| Chemical compound, drug | Glutathione-reduced | Sigma-Aldrich | G4251 | |
| Chemical compound, drug | Bromophenol blue | Sigma-Aldrich | B8026 | |
| Chemical compound, drug | HEPES | Sigma Aldrich | H3784 | |
| Chemical compound, drug | ATP | Cell Signaling Technology | 9804 | |
| Chemical compound, drug | $\gamma-^{32}P$-ATP | Bhabha Atomic Research Centre, India | | |
| Chemical compound, drug | [$^3$H]leucine | Amersham Biosciences | TRK510 | |
| Chemical compound, drug | $NAD^+$ | Sigma Aldrich | NAD100-RO-Roche | |
| Equipment | VisualSonics high-frequency ultrasound system | Vevo 1100 | | |
| Equipment | SDS-PAGE Gel running apparatus | Bio-Rad | | |
| Equipment | Western blotting apparatus | Bio-Rad | | |
| Equipment | Scintillation counter | Beckman | | |
| Equipment | Chemiluminescence imager | Chemidoc Touch, Biorad, USA | | |
| Equipment | ThermoMixer C | Eppendorf | | |
| Equipment | Power-pack | Bio-Rad | | |
| Equipment | LSM 880 confocal microscope | Zeiss | | |
| Equipment | Tissue-culture ware | Eppendorf | | |
| Software, algorithm | GraphPad Prism 5 | GraphPad Software | | |
| Software, algorithm | QuickChange Primer Design | Agilent Genomics | | |
| Software, algorithm | ImageJ | National Institutes of Health | | |
| Software, algorithm | ZEN 5 | Zeiss | | |
| Software, algorithm | Image Lab | Bio-Rad | | |
| Software, algorithm | Mascot data explorer software | Matrix Science, London, United Kingdom | | |
| Software, algorithm | Scaffold_2.1.03 | Proteome Software, Inc., Portland, OR | | |
| Software, algorithm | Swiss-model tool | ExPASy web server | PMID:24782522 | |

*Continued*

| Reagent type (species) or resources | Designation | Source or reference | Identifiers | Additional information |
|---|---|---|---|---|
| Software, algorithm | UCSF Chimera software package | Resource for Biocomputing, Visualization, and Informatics, NHI | | PMID:15264254 |
| Software, algorithm | GROMACS simulation package, version 5.0.4 | | | |
| Software, algorithm | PyTMs plugin of PyMOL | | | https://doi.org/10.1186/s12859-014-0370-6 |
| Miscellaneous | Osmotic Minipumps | ALZET | Models 2002, 2001 | PMID: 19652361 |
| Miscellaneous | Cover-slip 18 mm | Blue Star Slides | | |
| Miscellaneous | PVDF membrane Amersham Hybond P | GE Healthcare | 10600023 | |
| Miscellaneous | Nitrocellulose paper | Biorad | | |
| Miscellaneous | Cell culture wares | Eppendorf | | |
| Miscellaneous | Sigma cell scraper | Sigma-Aldrich | SIAL0010 | |

## Animal experiments

All animal experiments were performed with the approval of Institutional animal ethics committee of Indian Institute of Science, Bengaluru, India. All the animal experiments were carried out as per the strict accordance with the recommendations of the Committee for the Purpose of Control and Supervision of Experiments on Animals (CPCSEA), Government of India. The protocols were approved by the Institutional Animal Ethics Committee of the Indian Institute of Science (Permit Numbers: 559/2017, 568/2017, 376/2014). Mice were sacrificed using $CO_2$ before harvesting and every effort was made to minimize suffering. 129/SvJ mice (renamed as 129 $\times$ 1/SvJ, JAX stock # 000691) and Sirt2 knockout mice (JAX stock #012772 - B6.129-Sirt2 < tm1.1Fwa>/J) were procured from the Jackson Laboratories, USA. Mice were housed in individually ventilated cages (IVC) under 12 hr light/dark cycle in the clean air facility of Central Animal Facility, Indian Institute of Science. Chow diet and water were given ad libitum to animals.

## Induction of cardiac hypertrophy in mice

Cardiac hypertrophy was induced in 8 weeks old 129/Sv mice by surgically implanting Isoproterenol (ISO)-filled osmotic minipumps (models 2002, 2001; ALZET), into the peritoneal cavity of the mice. ISO (Sigma-Aldrich) was dissolved in buffer containing 150 mM NaCl and 1 mM acetic acid and delivered chronically at the dose of 10 mg/kg/day for 7 days. Control mice underwent similar procedure, except that the pumps were filled with vehicle solution containing 150 mM NaCl and 1 mM acetic acid. The induction of hypertrophy was noninvasively assessed by use of echocardiography. Mice were sacrificed, the heart was quickly collected and soaked in ice cold PBS. Heart was squeezed to remove excess blood and PBS, and heart weight was measured by a weighing balance. Tibia length of mice was measured by a Vernier caliper. The heart was sectioned into smaller pieces, immediately snap-frozen in liquid nitrogen and stored at −80° till further processed.

## Echocardiography of mice

Mice were anaesthetized by continuously infusing ~1% Isoflurane via nasal cone. A commercial topical depilatory was used to remove chest hair of mice. Body temperature of the mice was maintained by a heated imaging platform. Electrocardiogram leads were attached to limb of mice for gating. FUJIFILM VisualSonic Vevo 1100 high-frequency ultrasound equipped with 30 MHz high-frequency transducer was used to image the animals in the left lateral decubitus position. Two-dimensional echocardiographic images were recorded in parasternal long- and short-axis projections, with guided M-mode recordings at the midventricular level in both views. Left ventricular cavity size and left ventricular wall thickness were measured in at least three beats from each projection and averaged for data analysis. Left ventricular fractional shortening was calculated from the M-mode measurements by an in-built software.

## Cell lines, plasmids and transfection

HeLa and HEK 293 cell lines were purchased from ATCC. The identity of the cell lines has been authenticated by short tandem repeat (STR) analysis by the supplier. These cells were tested myco-plasma free by PCR-based screening in the laboratory. GSK3β-Knock-out (GSK3β-KO) mouse embryonic fibroblasts were kindly provided by James Woodgett, Mount Sinai Hospital, Toronto, Canada. HeLa, HEK 293 and GSK3β-KO embryonic fibroblasts were cultured in Dulbecco's Modified Eagle Medium (DMEM) supplemented with 10% fetal bovine serum (FBS), 100 units/ml penicillin and 100 μg/ml streptomycin at 37°C and 5% $CO_2$. Refer Key resources table for source and description of plasmids used. Site directed mutagenesis was carried out as per previously described protocol (Zheng et al., 2004). QuickChange Primer Design tool was used to design primers for site-directed mutagenesis of Sirt2 and Gsk3b. The site-directed mutation was verified by sequencing (SciGenom). Cells were transfected with plasmids harboring gene of interest using Lipofectamine 2000 transfection reagent as per manufacturer's protocols. Briefly, plasmid encoding control, or the desired construct were diluted in serum-free media and incubated for 5 min. Similarly, Lipofectamine was diluted in serum-free media, and incubated for 5 min. Mixtures were vortexed and centrifuged. Post-incubation, diluted plasmid and lipofectamine were pooled together, vortexed, centrifuged, and incubated at room temperature for 30 min. Equal volume of the mixture was added to the cells in serum-free media for 6 hr.

## Culture of cardiomyocytes

Primary rat neonatal cardiomyocytes were isolated from 1- or 2-day-old Wistar rats as described earlier (Jain et al., 2017). Rat pups were anaesthetized using 1–2% isoflurane and were sacrificed by decapitation. The excised hearts were placed in sterile ice-cold PBS containing D-glucose (0.01 M), minced into small pieces and further enzymatically digested by mixture of 0.2% trypsin, Collagenase Type II (0.4 mg/ml) and 0.01M D-glucose containing PBS in a 1.5 ml microcentrifuge tube. Approximately 65–75 μl of the digestion mixture was used per heart. Enzymatic digestion was carried out for 5 min in 37°C with shaking at 250 rpm. The unwanted erythrocytes and the debris were discarded from the first round of digestion. A total of eight to ten rounds of digestion was carried out and the supernatant containing a single-cell suspension from each digestion was collected in a 15-ml falcon tube containing 100% horse serum. The cell suspension was maintained at 37°C throughout the digestion process. Isolated neonatal rat cardiomyocytes were pre-plated on uncoated tissue culture plates to remove the adherent fibroblasts. After 1 hr of pre-plating, the non-adherent cell population enriched with cardiomyocytes were collected and centrifuged at 1000 rpm for 10 min. The supernatant was discarded, and cell pellet suspended in high-glucose DMEM supplemented with 10% FBS, 100 units/ml penicillin and 100 μg/ml streptomycin. Cardiomyocytes were seeded onto gelatin (0.2% w/v)-coated sterile tissue culture dishes for further experiments.

## Adenovirus infection

Adenovirus vectors encoding SIRT2 or p300 were purchased from Vector Biolabs. Adenovirus vectors synthesizing shRNA against Luc or p300 were kindly gifted by B. Thimmapaya, Northwestern University, Chicago, IL. After 36 hr of plating, cardiomyocytes were infected with adenoviruses; Ad-null, Ad-GFP, Ad-p300, Ad-SIRT2, Ad-Luc-shRNA or Ad-p300-shRNA at multiplicity of infection (MOI) of 10. All adenovirus infection experiments were carried out in high-glucose DMEM supplemented with 10% FBS, 100 units/ml penicillin and 100 μg/ml streptomycin.

## siRNA transfection

Smart pool SIRT2 siRNA and non-targeting siRNA were purchased from Dharmacon. For the siRNA experiments, cardiomyocytes were transfected with 100 nM SMARTpool siRNA targeting rat SIRT2 using Lipofectamine RNAiMAX Transfection Reagent as per manufacturer's protocol. Non-targeting siRNA was used as control. Briefly, non-targeting siRNA or SMARTpool siRNA targeting rat SIRT2 were diluted in serum-free media and incubated for 5 min. Similarly, Lipofectamine RNAiMAX Transfection Reagent was diluted in serum-free media and incubated for 5 min. Mixtures were vortexed and centrifuged. Post-incubation, diluted plasmid and lipofectamine were pooled together, vortexed, centrifuged, and incubated at room temperature for 30 min. Equal volume of the mixture was added to the cells in serum-free media for 6 hr.

## Cardiomyocyte hypertrophy experiments

Neonatal rat cardiomyocytes were exposed to ISO for inducing cardiomyocyte hypertrophy. Changes in morphology of cardiomyocytes were assessed by observing the sarcomere reorganization by immunostaining of cardiomyocytes with antibodies specific for $\alpha$-actinin or myomesin. Image J software was used to measure the cardiomyocyte surface area in actinin or myomesin-positive cells in an experimental group. To confirm cardiomyocyte hypertrophy, perinuclear expression of ANP was assessed by confocal microscopy. Briefly, cells were fixed with 3.7% formaldehyde for 15 min at room temperature. Fixed cells were washed thrice with PBS and permeabilized for 5 min with PBS containing 0.25% Triton X-100. After washing thrice with PBS, fixed cells were blocked with 5% BSA prepared in PBST (PBS with 1% Tween 20) containing glycine (22.52 mg/ml) for 1 hr. Following blocking, cells were incubated with anti-ANP (Cloud-clone corp. or abcam) and myomesin (DSHB) or actinin (abcam) antibodies prepared in 1% BSA containing PBST at 4°C overnight. Further, cells were incubated with secondary antibody conjugated with Alexa fluor 488 and Alexa fluor 546 at room temperature for 1 hr. After washing thrice with PBS for 2 min, cells were incubated with Hoechst 33342 for 10 min to stain nucleus. After nuclear staining, cells were washed thrice with PBS. and mounted on slides using Fluoromount G. Zeiss LSM 880 confocal microscope was used for image acquisition and ZEN-Black software was used for image analysis.

## [$^3$H]-Leucine incorporation assay

Leucine incorporation assay was performed as previously described (*Pillai et al., 2015*). Briefly, cardiomyocytes treated with either vehicle or ISO were incubated with [$^3$H]-leucine (1.0 mCi/ml, 163 Ci/mmol specific activity, Amersham Biosciences) in leucine-free minimal essential medium (Invitrogen) for 24 hr. Cells were then washed with phosphate-buffered saline, and 10% trichloroacetic acid was added to the cells to precipitate total proteins. The resultant protein pellet was solubilized using 0.2 N NaOH and further diluted with one-sixth volume of scintillation fluid. The radioactivity was measured in a scintillation counter and the values were normalized with DNA content measured by the Qubit dsDNA HS assay kit (Thermo Fisher Scientific).

## Non-radioactive SUnSET assay

To measure in-vitro protein synthesis, Surface sensing of translation (SUnSET), a non-radioactive method to monitor protein synthesis was employed (*Schmidt et al., 2009*). Cardiomyocytes seeded on six-well plates were pulsed with puromycin (1 µM) for 30 min prior to harvesting. Cells were washed twice with ice-cold PBS and lysed. Bradford assay was performed for protein quantification and 80 µg of protein was boiled in Laemmli Sample Buffer (Bio-Rad) supplemented with 5% β-mercaptoethanol for 5 min at 96°C. SDS-PAGE was performed, and the proteins were then transferred onto a 0.45 µm PVDF membrane (Amersham Hybond P, GE) by western blotting for 16 hr at 25V at 4°C. After 1 hr of blocking with 5% non-fat milk at room temperature, the membrane was incubated with anti-puromycin antibody (DSHB, University of Iowa) overnight at 4°C. Membrane was washed thrice with 1X TBST and incubated with anti-mouse secondary antibody for 1 hr at RT. Blot was washed thrice with 1X TBST and chemiluminescent signals were captured using BioRad Clarity ECL western Blotting Substrate in a chemiluminescence imager (Chemidoc Touch, Biorad).

## Inhibitor experiments

Anacardic acid was dissolved in peanut oil and injected into the peritoneal cavity of mice at a dose of 5 mg/kg/day for 10 days. LiCl (Sigma, Cat. No. 203637) or GSK3 inhibitor X (Cat. No. 361551) were used to inhibit the endogenous activity of GSK3 isoforms in cardiomyocytes. For inhibition of SIRT2 activity, cardiomyocytes were treated with either vehicle or AGK2, a SIRT2 inhibitor in high-glucose DMEM with 100 units/ml penicillin and 100 µg/ml streptomycin after 36 hr of seeding.

## Cells and tissue harvesting

Cells were washed twice with ice-cold phosphate buffered saline (PBS, 1X) before harvesting. Cells were lysed in ice-cold lysis buffer [20 mM Tris-Cl, pH 7.4, 150 mM NaCl, 1% Triton X-100, 1 mM EDTA, 1 mM EGTA, 2.5 mM sodium pyrophosphate, 1 mM Na$_3$VO$_4$, 1 mM PMSF, 1X protease inhibitor cocktail (Roche)]. Heart tissue was homogenized in lysis buffer [50 mM Tris-Cl, pH 7.4 150 mM NaCl, 1% Triton-X-100, 0.5% Sodium deoxycholate, 0.1% SDS, 1 mM EDTA, 10 mM Sodium fluoride,

2.5 mM sodium pyrophosphate, 1 mM Na$_3$VO4, 1 mM PMSF, 1X protease inhibitor cocktail (Roche)]. Homogenates were centrifuged at 12,000 rpm at 4°C for 10 min and the supernatant was collected in fresh micro-centrifuge tubes for western blotting.

## Western blotting

Western blotting was performed as per the standard protocols. Protein quantification was done by Bradford reagent using a spectrophotometer. Cell or tissue lysates was mixed with laemmli buffer (2X, Bio-rad) supplemented with 5% β-mercaptoethanol in 1:1 ratio. Samples were boiled for 5 min at 95°C for 5 min. SDS-PAGE was performed and the proteins were transferred to PVDF membrane by cold transfer (25V at 4° overnight). Protein-bound membrane was blocked by a solution of 5% non-fat dried milk in TBST buffer (25 mM Tris-HCl, pH 7.5, 150 mM NaCl, 0.05% Tween 20) for 1 hr at RT. Membrane was washed thrice with TBST buffer and probed with primary antibody at 4°C overnight. Membrane was further washed thrice with TBST buffer and probed with HRP-conjugated secondary antibody at RT for 1 hr. Membrane was washed thrice with TBST and the chemiluminescent signals recorded using either Clarity ECL western Blotting Substrate (BioRad) or SuperSignal West Pico chemiluminescent Substrate (ThermoScientific) in a chemiluminescence imager (Chemidoc Touch, Biorad).

## Immunoprecipitation assays

Immunoprecipitation assay was performed as per the standard protocols. For immunoprecipitation, 0.5–1 mg of protein from lysate was incubated with 2 μg of appropriate antibody or control IgG antibody overnight. Protein A/G-conjugated agarose beads were used to capture the immune complexes. After a brief centrifugation, supernatant was discarded, and beads were washed thrice with ice-cold PBS (1X) 1000 rpm at 4°C for 1 min. The immunoprecipitated protein was resolved by SDS-PAGE after boiling the beads at 95°C for 5 min in 2X laemmli buffer (Bio-rad) and transferred to a PVDF membrane (GE, Cat#10600023). The membrane was blocked by a solution of 5% non-fat dried milk in TBST buffer (25 mM Tris-HCl, pH 7.5, 150 mM NaCl, 0.05% Tween 20) for 1 hr at RT. Membrane was washed thrice with TBST buffer and probed with primary antibody at 4° for overnight. After washing thrice with TBST buffer and the membrane was probed with HRP-conjugated secondary antibody (Clean-Blot IP Detection Reagent, Thermo Scientific or Mouse monoclonal SB62a Anti-Rabbit IgG light chain (HRP) (ab99697) prepared in 1% milk at RT for 1 hr. Blot was washed thrice with TBST (1X) and chemiluminescent signals were captured using Clarity ECL Western Blotting Substrate (BioRad) in a chemiluminescence imager (Chemidoc Touch, Biorad).

## Co-localization experiments

To study the co-localization of p300 with GSK3β, confocal microscopy was performed in HEK 293 cells using antibodies specific to p300 and GSK3β. Similarly, to test the localization of HA-tagged GSK3β-WT or GSK3β-K183R or GSK3β-K183Q mutants, plasmid encoding HA-tagged GSK3β-WT or GSK3β-K183R or GSK3β-K183Q mutants were transiently overexpressed in GSK3β deficient mouse embryonic fibroblasts by Lipofectamine 2000 as per manufacturers protocols. After 48 hr of transfection, confocal microscopy was performed by an HA antibody to localize WT and mutants of GSK3β. To test the effect of AGK2 on GSK3β localization, confocal microscopy was performed with an antibody specific to GSK3β and/or anti-MnSOD.

For performing confocal microscopy, cells were washed twice with PBS and fixed with 3.7% formaldehyde at room temperature for 15 min. Fixed cells were washed thrice with PBS and incubated for 5 min with 0.25% Triton X-100 prepared in PBS. After washing thrice with PBS, cells were blocked with 5% BSA prepared in PBST (PBS with 1% Tween 20) containing glycine (22.52 mg/ml) for 1 hr. Following blocking, cells were incubated with respective primary antibodies prepared in 1% BSA in PBST at 4°C overnight. Further, cells were incubated with secondary antibody conjugated with Alexa fluor 488 and/or Alexa fluor 546. After 1 hr of incubation at room temperature, antibody was discarded and washed thrice with PBS. Hoechst 33342 prepared in PBS was added for 10 min to stain nucleus. After nuclear staining, cells were washed thrice with PBS and Fluoromount G was used to mount slides. Confocal images were captured by Zeiss LSM 880 confocal microscope.

## GST-GSK3β purification

Recombinant GST-GSK3β encoding expression plasmid was transformed into competent *E.coli* BL21 (DE3) cells. Culture from single colony was grown in LB medium supplemented with ampicillin (100 µg/ml) at 37°C in orbital shaker incubator till mid log phase. 50 µM IPTG was added to the culture and incubated at 18°C for 16 hr. The culture was centrifuged at 5000 rpm at 4°C for 20 min, and the pellet resuspended in binding buffer (50 mM Tris-Cl, pH 8.0, 150 mM NaCl, 1 mM PMSF (Sigma)). The resuspended cell pellet was lysed by sonication and centrifuged at 17,000 rpm at 4°C for 30 min. After centrifugation, supernatant was collected and incubated with Glutathione Sepharose 4B beads. Beads were washed thrice with wash buffer (50 mM Tris-Cl, pH 8.0, 150 mM NaCl, 0.5% Triton-X 100), GST-GSK3β eluted with elution buffer (20 mM glutathione-SH), and stored in sterile glycerol (30%) at −80°C.

## Purification of HIS-GSK3β and its mutants

Recombinant HIS-GSK3β or its mutants HIS-GSK3β-K183R, HIS-GSK3β-K183Q encoding expression plasmid were transformed into competent *E. coli* BL21 (DE3) cells. Culture from single colony was grown in LB medium supplemented with ampicillin (100 µg/ml) at 37°C in orbital shaker incubator till mid log phase. 50 µM IPTG was added to the culture and incubated at 18°C for 18 hr. The culture was centrifuged at 5000 rpm at 4°C for 20 min, and cell pellet was lysed by sonication in lysis buffer (20 mM Tris-Cl, pH 7.4, 500 mM NaCl, 20 µM imidazole, 1% Triton-X 100). Sonicated lysate was centrifuged at 17,000 rpm at 4°C for 30 min. Supernatant fraction was passed through pre-washed Ni-NTA column three times. Column was washed with five-column volume wash buffer (50 mM Tris-Cl (pH 7.4), 500 mM NaCl, 20 mM imidazole, 0.5% Triton-X-100). Protein was eluted by elution buffer (50 mM Tris-Cl (pH 7.4), 150 mM NaCl, 500 mM imidazole, 0.5% Triton-X-100) and stored in sterile glycerol (30%) at −80°C.

## GSK3 acetylation assay

His-GSK3 and HA-GSK3 was incubated with 1 µg of recombinant p300 acetyltransferase (Millipore # 2273152) in 50 µl of HAT buffer (50 mM Tris-Cl, pH 8.0, 1 mM EDTA, 10 mM Na-butyrate, 5 mM DTT, 10 mM NaCl and 10% glycerol) supplemented with 100 µM acetyl CoA. Samples were incubated at 30°C for 2 hr. Beads were washed thrice with TBS (25 mM Tris-HCl, pH 7.5, 150 mM NaCl) and the protein eluted by adding 50 µl of sample buffer (125 mM Tris-Cl, pH 6.8, with 4% SDS, 20% (v/v) glycerol, and 0.004% bromophenol blue). Samples were heated for 5 min at 95°C followed by centrifugation at 13,400 g, 30 s, and supernatant subjected to SDS-PAGE. GSK3β acetylation was detected by a pan anti-acetyl Lysine antibody (Cell Signaling #9681).

## GSK3 deacetylation assay

Plasmid encoding flag-tagged WT SIRT2 or SIRT2-H187Y was overexpressed in 293 cells by transfection with lipofectamine 2000. Cells were harvested after washing with ice-cold PBS and lysed in lysis buffer [50 mM Tris-Cl, pH 7.4, 150 mM NaCl, 1 mM EDTA, 1% Triton X-100, 1 mM PMSF and protease inhibitor cocktail (Sigma Aldrich)] followed by vortexing for 15 s at 5 min interval for 4–5 times. Cell homogenates were centrifuged at 12,000 rpm at 4°C for 10 min and the supernatant was collected. 500 µg of total protein was incubated with agarose beads conjugated to anti-Flag antibody (Sigma A2220) and kept for end-over-end mixing for 2 hr at 4°C. Beads were washed thrice with TBS (25 mM Tris-HCl, pH 7.5, 150 mM NaCl) and the SIRT2 protein eluted. His-tagged or HA-tagged acetylated GSK3 was deacetylated by recombinant SIRT2 in a deacetylation buffer (250 mM Tris-Cl, pH 9.0, 20 mM MgCl$_2$, 250 mM NaCl, 2.5 mM DTT, 5 mM NAD$^+$, 2.5 µM TSA) for 2 hr at 30°C. The acetylation of GSK3 isoforms was analyzed by western blotting.

## ATP binding assay

His-tagged wild type and mutants of GSK3 or His-tagged acetylated and deacetylated GSK3 bound to Ni-NTA beads were incubated with [γ $^{32}$P] ATP (2 µCi) at 30°C for 10 min in a binding buffer containing 20 mM HEPES pH 7.5, 50 mM NaCl, 10 mM MgCl$_2$, 2 mM CaCl$_2$, 200 µM ATP in a final volume of 50 µL. After 30 min, the beads were washed 5 times with 50 mM Tris (pH 7.5) and subjected to scintillation counting using a counter (Beckman).

## Activity assay for GSK3 isoforms

Commercially available GSK3 activity assay kit (CS0990; Sigma) was used for performing GSK3 activity assay as per manufacturer's instructions. Briefly, HA-tagged GSK3β or GSK3α was immunoprecipitated using a specific anti-HA antibody conjugated agarose beads (Sigma-Aldrich). Similarly, endogenous GSK3β was immunoprecipitated from heart tissue lysates or cell lysates by a specific anti-GSK3β antibody bound protein A/G affinity gel. The immunoprecipitated kinase was incubated with $\gamma-^{32}$P-ATP and the incorporation of $^{32}$P into glycogen synthase peptide, which contains specific phosphorylation residue of GSK3 was measured.

## Activity assay for acetylated and deacetylated-GSK3 isoforms

HA-GSK3β or HA-GSK3α was overexpressed in HeLa cells by transfection of the respective plasmid encoding pcDNA3-HA-GSK3β or pcDNA3-HA-GSK3α. HeLa cell lysates were incubated with monoclonal Anti-HA-agarose antibody-conjugated agarose beads for 2 hr at 4°C with end-over-end mixing. HA-GSK3β or HA-GSK3α bound to beads was incubated with recombinant p300 (Millipore, Temecula, CA, 1 μg) in 50 μl of HAT buffer (50 mM Tris-Cl, pH 8.0, 1 mM EDTA, 10 mM Na-butyrate, 5 mM DTT, 10 mM NaCl and 10% glycerol) supplemented with 100 μM acetyl CoA at 30°C for 60 min. Beads were washed thrice with TBS (25 mM Tris-HCl, pH 7.5, 150 mM NaCl) and further deacetylated by either SIRT2 or SIRT2-catalytic mutant in a deacetylation buffer (250 mM Tris-Cl, pH 9.0, 20 mM MgCl₂, 250 mM NaCl, 2.5 mM DTT, 5 mM NAD⁺, 2.5 μM TSA) for 60 min at 30°C. The acetylated and deacetylated HA-GSK3β or HA-GSK3α was incubated with $\gamma-^{32}$P-ATP in a kinase buffer and the incorporation of $^{32}$P into glycogen synthase peptide, containing specific phosphorylation residues of GSK3 was measured as per the protocols of GSK3 activity assay kit (CS0990; Sigma).

## Mass spectrometry analysis

GSK3 isoforms were resolved by SDS-PAGE, stained with colloidal Coomassie and the desired protein bands were excised from the gel by the use of a razor blade and divided into ~1 mm$^3$ pieces. The gel pieces containing protein were destained using 100 mM ammonium bicarbonate (pH 8.9) in 50% acetonitrile. The destained bands were further treated with 100 μl of 50 mM ammonium bicarbonate (pH 8.0) and 10 μl of 10 mM TCEP [Tris (2-carboxyethyl)phosphine HCl] at 37°C for 30 min. Subsequently, protein digestion was carried out by 1:50 sequencing-grade trypsin in 50 mM ammonium bicarbonate (pH 7.5) solution. Digested peptide samples were desalted in a C₈ OptiPak column (Optimize Technologies) and then analyzed by liquid chromatography-electrospray tandem mass spectrometry (LC-ESI/MS/MS) on a Thermo LTQ Orbitrap Hybrid FT mass spectrometer. Reflectrom mode was used to acquire positive-ion mass spectra. Ions selected for MS/MS were subsequently placed on an exclusion list using an isolation width of 1.6 Da, a low-mass exclusion of 0.8 Da, and a high-mass exclusion of 0.8 Da. Tandem mass spectra were extracted by Readw.exe version 3.0. All MS/MS samples were analyzed using Mascot data explorer software (Matrix Science, London, United Kingdom). The Scaffold (version Scaffold_2.1.03; Proteome Software, Inc., Portland, OR) was used to validate MS/MS-based peptide and protein identifications.

## Modeling of GSK3α

Swiss-model tool was used to generate the homology model of GSK3α (*Biasini et al., 2014*). The sequence-based template search results showed ~82% MS sequence identity with the crystal structure of GSK3β (PDB ID 1PYX). Further, the crystal structure with PDB ID 1PYX was used to generate the final model of GSK3α; the final model of GSK3α encompasses the residues 98 to 448. UCSF Chimera software package was used for visualization and generation of the final images (*Pettersen et al., 2004*).

## Modeling of ac183 GSK3β molecular dynamics simulations and analysis of trajectories

Models of GSK3β protein were generated from the crystal structure (PDB ID 4NM0 A). Computer aided acetylated lysine (acK183) mutant was generated over the crystal structure (PDB ID 4NM0 A), using the PyTMs plugin of PyMOL (*Warnecke et al., 2014*). UCSF Chimera software package was used for visualization and generation of the final images (*Pettersen et al., 2004*).

MD Simulations on the initial models of GSK3β wild-type and acetylated GSK3β were carried out with the GROMACS simulation package, version 5.0.4. The systems were parameterized using CHARMM36 force field and TIP3P rigid water model (*Jorgensen et al., 1983*). The atomic charges for terminal acetyl group available in the CHARMM36 force field were used to build the parameters for side-chain acetylated lysine. Each model was immersed in a dodecahedron box containing TIP3P water with 10 Å distance between the box surface and the protein atoms. Charge neutralization for the systems was achieved by adding $Na^+$ and $Cl^-$ counter ions. Steepest descent energy minimization was used until the system converged with Fmax no greater than 1000 kJ $mol^{-1}$ $nm^{-1}$. Equilibration was performed for 600ps under NVT and for 1200ps under NPT ensemble, while restraining the protein atomic positions. The coupling constant for the temperature of the bath was set to 0.1ps using v-rescale thermostat with a constant temperature of 300K. Berendsen thermostat was used to maintain the pressure a 1 bar with a coupling constant of 1ps. The electrostatic interactions were evaluated using the Particle Mesh Ewald (PME) method (*Darden et al., 1993*). Production run totaling to 100 ns each for the GSK3β wild-type and GSK3β acK183 systems were performed on the equilibrated systems using leap-frog algorithm.

GROMACS tools were used to calculate the root mean square deviation (RMSD) of Cα atoms, root mean square deviation (RMSD) of ADP nucleotide, resultant root mean square fluctuations (RMSF) of Cα atoms on the five MD trajectories of 20 ns each for both the wild type and acK183 mutant of GSK3β. UCSF Chimera software tool was used for visualization and calculation of distances between the atoms (*Pettersen et al., 2004*).

### Quantification and statistical analysis

Statistical analysis and graph preparation was done by Graph-pad prism version 6.04. t-test was used for pair-wise comparisons. One-way ANOVA, and two-way ANOVA were used for multiple comparisons. ZEN-Black software was used for confocal image analysis and ImageJ was used for quantification. Densitometric analysis was performed using ImageJ. Western blotting images were processed by using Image-lab software (Bio-Rad).

## Acknowledgements

We thank James Woodgett, Mount Sinai Hospital, Toronto, Canada for providing GSK3β-KO mouse embryonic fibroblasts. We thank B Thimmapaya, Northwestern University, Chicago, IL, USA for providing adenovirus vectors synthesizing shRNA against p300. We thank Naren Ramanan, Centre for Neuroscience, Indian Institute of Science for kindly sharing the GSK3 inhibitor X. NRS is supported by the Ramalingaswami Re-entry Fellowship and the Innovative Young Biotechnologist Award (IYBA) from the Department of Biotechnology, Government of India. SM is supported by the SERB-National Post-Doctoral Fellowship (N-PDF). NRS laboratory is supported by research funding from the Department of Science and Technology Extra Mural Research Funding (EMR/2014/000065), The Department of Biotechnology Extramural Research Grants (BRB/10/1294/2014, MED/30/1454/2014), and The Council for Scientific and Industrial Research extramural research support (37 (1646)/15/EMR-II).

## Additional information

### Funding

| Funder | Grant reference number | Author |
|---|---|---|
| Department of Biotechnology, Ministry of Science and Technology | BRB/10/1294/2014 | Nagalingam R Sundaresan |
| Department of Biotechnology, Ministry of Science and Technology | MED/30/1454/2014 | Nagalingam R Sundaresan |
| Department of Biotechnology, Ministry of Science and Technology | IYBA Award | Nagalingam R Sundaresan |

| Department of Biotechnology, Ministry of Science and Technology | Ramalingaswami fellowship | Nagalingam R Sundaresan |
|---|---|---|
| Department of Science and Technology, Ministry of Science and Technology | EMR/2014/000065 | Nagalingam R Sundaresan |
| Council of Scientific and Industrial Research | 37(1646)/15/EMR-II | Nagalingam R Sundaresan |
| Department of Science and Technology, Ministry of Science and Technology | N-PDF | Sangeeta Maity |

The funders had no role in study design, data collection and interpretation, or the decision to submit the work for publication.

### Author contributions

Mohsen Sarikhani, Resources, Data curation, Formal analysis, Methodology, Writing—review and editing; Sneha Mishra, Data curation, Formal analysis, Methodology, Writing—original draft, Writing—review and editing; Sangeeta Maity, Data curation, Formal analysis, Validation, Methodology, Writing—review and editing; Chaithanya Kotyada, Data curation, Software, Formal analysis, Writing—original draft; Donald Wolfgeher, Data curation, Formal analysis; Mahesh P Gupta, Resources, Supervision; Mahavir Singh, Resources, Supervision, Writing—original draft, Writing—review and editing; Nagalingam R Sundaresan, Conceptualization, Supervision, Funding acquisition, Investigation, Methodology, Project administration, Writing—review and editing

### Author ORCIDs

Sneha Mishra (iD) http://orcid.org/0000-0001-5892-6816
Nagalingam R Sundaresan (iD) http://orcid.org/0000-0003-1770-5616

### Ethics

Animal experimentation: All animal experiments were performed with the approval of Institutional animal ethics committee of Indian institute of science, Bengaluru, India. All the animal experiments were carried out as per the strict accordance with the recommendations of the Committee for the Purpose of Control and Supervision of Experiments on Animals (CPCSEA), Government of India. The protocols were approved by the Institutional Animal Ethics Committee of the Indian Institute of Science (Permit Numbers: 559/2017, 568/2017, 376/2014 ). Mice were sacrificed using $CO_2$ before harvesting and every effort was made to minimize suffering.

### Decision letter and Author response

Decision letter https://doi.org/10.7554/eLife.32952.019
Author response https://doi.org/10.7554/eLife.32952.020

## Additional files

### Supplementary files

• Transparent reporting form
DOI: https://doi.org/10.7554/eLife.32952.016

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
