## [Decision Letter]

Thank you for submitting your article "SIRT2 deacetylase regulates GSK3β activity independent of inhibitory phosphorylation" for consideration by *eLife*. Your article has been reviewed by three peer reviewers, one of whom is a member of our Board of Reviewing Editors and the evaluation has been overseen by Jonathan Cooper as the Senior Editor. The following individual involved in review of your submission has agreed to reveal his identity: Bradley Doble (Reviewer #3).

The reviewers have discussed the reviews with one another and the Reviewing Editor has drafted this decision to help you prepare a revised submission.

Summary:

This is an interesting study that describes the regulation of GSK3 by acetylation. The authors report that acetylation on Lys183 reduces ATP binding and that the acetylation is mediated by p300 and that de-acteylation is mediated by SIRT2. The regulation by acetylation is stated to be independent of inhibitory phosphorylation. This is potentially important because GSK3 is a key signaling enzyme and the description of a new regulatory mechanism would be of wide interest. Several lines of evidence support the authors conclusions, but several points require clarification.

Essential revisions:

1) A major concern with this study is that the methods are very poorly described. Many experiments are not described at all (e.g. shRNA studies and antibodies used for immunofluorescence etc.) and some studies are described by citation to previous studies reported by the authors, but the citations are not helpful; for example, the methods states that rat cardiomyocytes were studied, but the citation is to the preparation of murine cardiomyocytes; the method used for ATP binding cites the use of His-tagged kinase and a kinase inhibitor – is this how the ATP binding was done? Moreover, the materials used for many of the experiments is not described – for example, the purified p300 used for binding assays (Figure 1) is not described, what was the source of acetylated GSK3 used for the de-acetylation assays (Figure 3), what was the source of the WT and mutant SIRT2 used for de-acetylation assays, etc. This is a major problem throughout the study and must be fixed for every experiment presented.

2) It has previously been shown that K189 in the Arabidopsis GSK3 orthologue BIN2 (equivalent to K205 in human GSK3β) is targeted by the deacetylase HDAC6 (Hao et al., 2016). In contrast to the regulation of K183 by SIRT2, deacetylation of K189 in BIN2 leads to a reduction in kinase activity. The authors should expand their discussion to indicate that distinct acetylation events within the kinase domain of GSK3 can have opposing effects on its kinase activity.

3) The relative role of SIRT2 is unclear. For example, do other SIRT enzymes play a role in GSK3 de-acetylation? Is the de-acetylation specific to SIRT2?

4) The relative role of p300 in acetylation of GSK3 is unclear. There is reduced acetylation in the p300 knockdown (Figure 1), but the effect is not robust. Is this because the knock-down was insufficient or because the potential role of p300 is redundant?

5) Both the mimetic and non-acetylatable mutants of K183 display reduced ATP binding and catalytic activity (Figure 2). This provides evidence that K183 is an important residue for ATP binding and GSKβ activity but is not a direct test of the effect of acetylation on ATP binding (as stated in subsection “Acetylation of GSK3β Influences ATP binding: Insights from Molecular Modeling and Molecular Dynamics Simulation of GSK3β WT and Acetylated K183 Mutant”). Similarly, the fact that the K183Q mutant decreases the activity of the S9A mutant (Figure 3) may simply reflect that this residue is critical for ATP binding, rather than proving that acetylation inhibits GSK3β activity independently of Ser9 phosphorylation (subsection “SIRT2 Modulates the Kinase Activity of GSK3β by Reversible Acetylation”). The discussion of these experiments should be more circumspect about the evidence for acetylation of K183 regulating ATP binding and the caveats should be clearly stated.

6) The ATP binding assay is unclear. The cited reference makes no mention of magnesium. Was the binding performed in the absence of magnesium?

7) The K183R mutant displays increased acetylation compared to wild-type GSK3β in the presence of active SIRT2 (Figure 3). What is the reason for this? Is there an increase of acetylation on other sites when K183 is mutated? This could have implications for the interpretation of the experiments where K183 mutants have been used. Quantification from multiple blots would also be helpful here.

8) The data from SIRT2-KO mice could be supported by the use of the SIRT2 inhibitor AGK2 on cardiomyocyte cultures. There also needs to be a control experiment included in Figure 3 that demonstrates that AGK2 is inhibiting SIRT2 under the conditions used. It has previously been reported that AGK2 decreases Ser9 phosphorylation of GSK3β in platelets (Moscardo et al., 2015) suggesting that the relationship between SIRT2 and Ser9 phosphorylation is context-dependent. This should be discussed.

9) The authors have previously reported the regulation of the PDK1/AKT pathway by acetylation. The authors should include discussion of this in the paper because the impression provided to the reader is that acetylation is only involved with direct regulation of GSK3 rather than also by an indirect mechanism via PDK1/AKT.

10) Presumably the sub-cellular localization of the sites of acetylation and de-acetylation of GSK3 are different (nuclear p300 and cytoplasmic SIRT2). The authors should comment on this.

11) The data directly linking GSK3 to the anti-hypertrophic effect of SIRT2 is mainly based on the use of LiCl as a GSK3 inhibitor (Figure 5). As LiCl is not specific for GSK3 but has other effects on cells, this data should be supported by the use of an additional GSK3 inhibitor or the GSK3β K183 mutants.

12) In Figure 5, the level of GSK3β acetylation does not correlate with GSK3β activity. In the presence of the p300 inhibitor, SIRT2-KO does not affect GSK3β acetylation but its activity is still significantly decreased. This should be commented on.

13) The Title and Abstract of the paper implies that the acetylation mechanism is specific to GSK3β, but GSK3α appears to share this regulatory mechanism. This conclusion should be confirmed by showing that GSK3α is inhibited by acetylation and reflected in the wording of a revised title.

[Editors' note: further revisions were requested prior to acceptance, as described below.]

Thank you for resubmitting your work entitled "SIRT2 deacetylase regulates the activity of GSK3 isoforms independent of inhibitory phosphorylation" for further consideration at *eLife*. Your revised article has been favorably evaluated by Jonathan Cooper (Senior editor) and Roger Davis (Reviewing editor).

The manuscript has been improved but there is a remaining issue that needs to be addressed before acceptance, as outlined below:

Original Point #1. The reviewers noted that the description of the reagents and the methods employed in this study was very poor. The revised manuscript has been improved. However, many problems remain. Based on the information provided, it would be very difficult for another investigator to repeat the studies that are described. The methods and reagents for each experiment must be described. The current Materials and methods section is not acceptable. There are many problems with the current text.

---

## [Author Response]

Essential revisions:1) A major concern with this study is that the methods are very poorly described. Many experiments are not described at all (e.g. shRNA studies and antibodies used for immunofluorescence etc.) and some studies are described by citation to previous studies reported by the authors, but the citations are not helpful; for example, the methods states that rat cardiomyocytes were studied, but the citation is to the preparation of murine cardiomyocytes; the method used for ATP binding cites the use of His-tagged kinase and a kinase inhibitor – is this how the ATP binding was done? Moreover, the materials used for many of the experiments is not described – for example, the purified p300 used for binding assays (Figure 1) is not described, what was the source of acetylated GSK3 used for the de-acetylation assays (Figure 3), what was the source of the WT and mutant SIRT2 used for de-acetylation assays, etc. This is a major problem throughout the study and must be fixed for every experiment presented.

We thank the editors and the reviewers for reviewing our manuscript. We have rewritten the Materials and methods section and incorporated all suggestions in the revised manuscript.

2) It has previously been shown that K189 in the Arabidopsis GSK3 orthologue BIN2 (equivalent to K205 in human GSK3β) is targeted by the deacetylase HDAC6 (Hao et al., 2016). In contrast to the regulation of K183 by SIRT2, deacetylation of K189 in BIN2 leads to a reduction in kinase activity. The authors should expand their discussion to indicate that distinct acetylation events within the kinase domain of GSK3 can have opposing effects on its kinase activity.

The residues K167 and K189 of BIN2 (referred as K167b and K189b from human GSK3β numbering) are analogous to the residues K183 and K205 of GSK3β respectively. We have generated a homology model of BIN2 using GSK3β as a template (Please see Author response image 1). It is clear that K183 (in GSK3β) is not equivalent to K189b (in BIN2). In a previous study, the acetylation of K189b in BIN2 was shown to increases its kinase activity (Hao et al., 2016). Our current study suggests that the acetylation of K183 decreases the GSK3β kinase activity by directly interfering with its adenine nucleotide binding (Figure 2). Since, the K189b resides in activation loop, away from the nucleotide-binding site, it would certainly follow a distinct mechanism in influencing the kinase activity of BIN2, when compared to K183 of GSK3β, which is situated in the nucleotide binding pocket. Therefore, distinct acetylation events within the kinase domain of GSK3 might have opposing effects on its kinase activity. We have included this point in the Discussion section.

**Author response image 1. respfig1:** Models of GSK3B and BIN2. (**A**) Surface representation of the crystal structure of GSK3B (PDB ID 4NM0, blue) and the homology model of BIN2 (salmon), highlighted are the analogous acetylation sites. (**B**) Overlay of the crystal structure of GSK3B (PDB ID 4NM0, blue) and the model of BIN2 (salmon). (**I**) Representation of analogous acetylation sites K183 (GSK3B) and K167 (BIN2), K205 (GSK3B) and K189 (BIN2). (II) Side-chain interaction between the residue K205 and N213 in crystal structure of GSK3B (PDB ID 4NM0).

3) The relative role of SIRT2 is unclear. For example, do other SIRT enzymes play a role in GSK3 de-acetylation? Is the de-acetylation specific to SIRT2?

GSK3β has been previously been to be deacetylated by SIRT3. Our previous study suggests that SIRT3 deacetylates GSK3β at K15 and activates GSK3β and thereby blocks TGF-β1 signalling and tissue fibrosis (Sundaresan et al., 2016). To further identify the sirtuins responsible for regulation of GSK3 activity, we overexpressed all the sirtuin isoforms and assessed the acetylation, phosphorylation and activity of GSK3β. Our results suggest that SIRT2 overexpression markedly reduced the acetylation of GSK3β, while increasing its activity against glycogen synthase. Interestingly, we do not observe any changes in the phosphorylation of Ser9 residue of GSK3β. However, we found increased phosphorylation of GSK3β at Tyr216. Consistent with our previous work (Sundaresan et al., 2016), we found SIRT3 is capable of deacetylating GSK3β and enhancing its catalytic activity (Figure 3—figure supplement 1). Please see subsection "SIRT2 modulates the kinase activity of GSK3β by reversible acetylation".

4) The relative role of p300 in acetylation of GSK3 is unclear. There is reduced acetylation in the p300 knockdown (Figure 1), but the effect is not robust. Is this because the knock-down was insufficient or because the potential role of p300 is redundant?

We agree with the reviewers that reduced acetylation of GSK3 in the p300 depleted cells is not robust. However, inhibition of SIRT2 by AGK2 (Figure 3—figure supplement 2) or depletion of SIRT2 (Figure 3), markedly enhances the acetylation of GSK3β. We believe that GSK3β acetylation might be regulated by multiple acetyltransferases including p300. Further studies are needed to identify potential acetyltransferases for GSK3β. We have discussed this point in the Results section (subsection “Acetyltransferase p300 Regulates the GSK3β Acetylation”).

5) Both the mimetic and non-acetylatable mutants of K183 display reduced ATP binding and catalytic activity (Figure 2). This provides evidence that K183 is an important residue for ATP binding and GSKβ activity but is not a direct test of the effect of acetylation on ATP binding (as stated in subsection “Acetylation of GSK3β Influences ATP binding: Insights from Molecular Modeling and Molecular Dynamics Simulation of GSK3β WT and Acetylated K183 Mutant”). Similarly, the fact that the K183Q mutant decreases the activity of the S9A mutant (Figure 3) may simply reflect that this residue is critical for ATP binding, rather than proving that acetylation inhibits GSK3β activity independently of Ser9 phosphorylation (subsection “SIRT2 Modulates the Kinase Activity of GSK3β by Reversible Acetylation “). The discussion of these experiments should be more circumspect about the evidence for acetylation of K183 regulating ATP binding and the caveats should be clearly stated.

We agree with reviewers and modified the statements in Results section to reflect the facts and caveats of these assay (Please see subsection “Acetylation of GSK3β Influences ATP binding: Insights from Molecular Modeling and Molecular Dynamics Simulation of GSK3β WT and Acetylated K183 Mutant”, subsection “SIRT2 Modulates the Kinase Activity of GSK3β by Reversible Acetylation”).

6) The ATP binding assay is unclear. The cited reference makes no mention of magnesium. Was the binding performed in the absence of magnesium?

We have included the detailed protocol used for ATP binding in the methods section. We have performed ATP binding assays in a binding buffer which has magnesium (Please see subsection “ATP binding assay”). Moreover, please note that magnesium is added along with ADP to the initial models prepared for MD simulations of both the wild-type and acetylated mutants of GSK3β.

7) The K183R mutant displays increased acetylation compared to wild-type GSK3β in the presence of active SIRT2 (Figure 3). What is the reason for this? Is there an increase of acetylation on other sites when K183 is mutated? This could have implications for the interpretation of the experiments where K183 mutants have been used. Quantification from multiple blots would also be helpful here.

Wild-type GSK3β acetylation reduces significantly, when SIRT2 is incubated in the presence of NAD^+^. Our result suggests that K183R mutant displays low basal acetylation, when compared to wild-type GSK3β. Further, GSK3β-K183R mutant is not deacetylated by SIRT2 even in the presence of NAD^+^. We have quantified our results from multiple experiments, and the results were included in the Figure 3. We believe that GSK3β-K183R mutant may not be an effective deacetylation target for SIRT2, though we do not have evidence to substantiate this claim. (Please see subsection “SIRT2 Modulates the Kinase Activity of GSK3β by Reversible Acetylation”)

8) The data from SIRT2-KO mice could be supported by the use of the SIRT2 inhibitor AGK2 on cardiomyocyte cultures. There also needs to be a control experiment included in Figure 3 that demonstrates that AGK2 is inhibiting SIRT2 under the conditions used. It has previously been reported that AGK2 decreases Ser9 phosphorylation of GSK3β in platelets (Moscardo et al., 2015) suggesting that the relationship between SIRT2 and Ser9 phosphorylation is context-dependent. This should be discussed.

As suggested by the reviewers, cardiomyocytes were treated with AGK2, a SIRT2 inhibitor to test phosphorylation and acetylation status of GSK3β (Figure 3—figure supplement 2). We have also tested acetylation of tubulin K40 residue by specific antibody to assess the SIRT2 activity (Figure 3—figure supplement 2). (Please see subsection “SIRT2 Modulates the Kinase Activity of GSK3β by Reversible Acetylation”)

9) The authors have previously reported the regulation of the PDK1/AKT pathway by acetylation. The authors should include discussion of this in the paper because the impression provided to the reader is that acetylation is only involved with direct regulation of GSK3 rather than also by an indirect mechanism via PDK1/AKT.

As suggested, we have discussed these points in the Discussion section.

10) Presumably the sub-cellular localization of the sites of acetylation and de-acetylation of GSK3 are different (nuclear p300 and cytoplasmic SIRT2). The authors should comment on this.

GSK3 is reported to be localised in different subcellular localisation viz, nucleus, cytoplasm, mitochondria. As reviewers pointed that the p300 is mostly nuclear protein and SIRT2 is majorly cytoplasmic protein. Since, GSK3 shuttles between cytoplasm and nucleus, it is possible that p300 and SIRT2 may function as acetyl transferase and deacetylase respectively, though they are localized in different subcellular compartments. (Discussion section).

11) The data directly linking GSK3 to the anti-hypertrophic effect of SIRT2 is mainly based on the use of LiCl as a GSK3 inhibitor (Figure 5). As LiCl is not specific for GSK3 but has other effects on cells, this data should be supported by the use of an additional GSK3 inhibitor or the GSK3β K183 mutants.

We have repeated the experiment using GSK3 specific inhibitor, GSK3-inhibitor X and the results were presented in Figure 5. (Please see subsection “GSK3 is Required for the Anti-Hypertrophic Role of SIRT2 deacetylase”).

12) In Figure 5, the level of GSK3β acetylation does not correlate with GSK3β activity. In the presence of the p300 inhibitor, SIRT2-KO does not affect GSK3β acetylation but its activity is still significantly decreased. This should be commented on.

We agree to reviewers that the change in acetylation status of GSK3β after p300 inhibitor treatment is not correlated well with the GSK3β activity on Glycogen synthase. It could be due to modifications of glycogen synthase in SIRT2-KO hearts. We also believe that glycogen synthase might have also been post-translationally modified in SIRT2-KO hearts. Another reason could be SIRT2-KO hearts might have increased expression of phosphatases that might regulate glycogen synthase phosphorylation. We have discussed these points in Results section (Please see subsection “GSK3 is Required for the Anti-Hypertrophic Role of SIRT2 deacetylase”).

13) The Title and Abstract of the paper implies that the acetylation mechanism is specific to GSK3β, but GSK3α appears to share this regulatory mechanism. This conclusion should be confirmed by showing that GSK3α is inhibited by acetylation and reflected in the wording of a revised title.

We agree to the reviewer that GSK3α is also inhibited by acetylation. We performed GSK3α activity assay, and the result has been incorporated in revised manuscript (Please see Figure 4; subsection “Molecular Modeling of GSK3α Suggests that the Acetylation of Residue K246 would have Similar Consequences as K183 of GSK3β”). We have also edited the Title and Abstract to reflect the findings of the manuscript, as suggested by the reviewers.

[Editors' note: further revisions were requested prior to acceptance, as described below.]

Original Point #1. The reviewers noted that the description of the reagents and the methods employed in this study was very poor. The revised manuscript has been improved. However, many problems remain. Based on the information provided, it would be very difficult for another investigator to repeat the studies that are described. The methods and reagents for each experiment must be described. The current Materials and methods section is not acceptable. There are many problems with the current text.

We thank the editors and the reviewers for the valuable suggestion. We have rewritten the Materials and methods section and figure legends to reflect the detailed protocols for each experiment.

We have also included a Key Resources Table in the methods section of the manuscript, which describes the source of all materials used in the current work.